# Contrastive and Non-Contrastive Self-Supervised Learning Recover Global and Local Spectral Embedding Methods

**Randall Balestriero**
Meta AI Research, FAIR
NYC, USA
rbalestriero@meta.com

**Yann LeCun**
Meta AI Research, FAIR, NYU
NYC, USA
ylecun@meta.com

## Abstract

Self-Supervised Learning (SSL) surmises that inputs and pairwise positive relationships are enough to learn meaningful representations. Although SSL has recently reached a milestone: outperforming supervised methods in many modalities. . . the theoretical foundations are limited, method-specific, and fail to provide principled design guidelines to practitioners. In this paper, we propose a unifying framework under the helm of spectral manifold learning to address those limitations. Through the course of this study, we will rigorously demonstrate that VICReg, SimCLR, BarlowTwins et al. correspond to eponymous spectral methods such as Laplacian Eigenmaps, Multidimensional Scaling et al. This unification will then allow us to obtain (i) the closed-form optimal representation for each method, (ii) the closed-form optimal network parameters in the linear regime for each method, (iii) the impact of the pairwise relations used during training on each of those quantities and on downstream classification task performances, and most importantly, (iv) the first theoretical bridge between contrastive and non-contrastive methods towards global and local spectral embedding methods respectively, hinting at the benefits and limitations of each. For example, (i) if the pairwise relation is aligned with the downstream task, any SSL method can be employed successfully and will recover the supervised method, but in the low data regime, SimCLR or VICReg with high invariance hyper-parameter should be preferred; (ii) if the pairwise relation is misaligned with the downstream task, BarlowTwins or VICReg with small invariance hyper-parameter should be preferred.

## 1 Introduction

Self-Supervised Learning (SSL) is one of the most promising method to learn data representations that generalize across downstream tasks. SSL places itself in-between supervised and unsupervised learning as it does not require labels but does require knowledge of what makes some samples semantically close to others. Hence, where unsupervised learning relies on a collection of inputs $(\boldsymbol{X})$, and supervised learning relies on inputs and outputs $(\boldsymbol{X}, \boldsymbol{Y})$, SSL relies on inputs and inter-sample relations $(\boldsymbol{X}, \boldsymbol{G})$ that indicate semantic similarity. The latter matrix $\boldsymbol{G}$ is often constructed by augmenting $\boldsymbol{X}$ through data-augmentations known to preserve input semantics [1–3] e.g. horizontal flip for an image, although recent methods have went away from Data-Augmentation (DA)by using videos from which consecutive frames can be seen as semantically equivalent [4–6].

Although SSL originated decades ago [7], recent advances have pushed SSL performances beyond expectations [8–10]. Due to those rapid empirical advances, an urgent need for a principled theoretical understanding of those methods has emerged [11, 12]. Studies in this direction often take one of the three following approaches: (i) studying the training dynamics and optimization landscapes of

36th Conference on Neural Information Processing Systems (NeurIPS 2022).

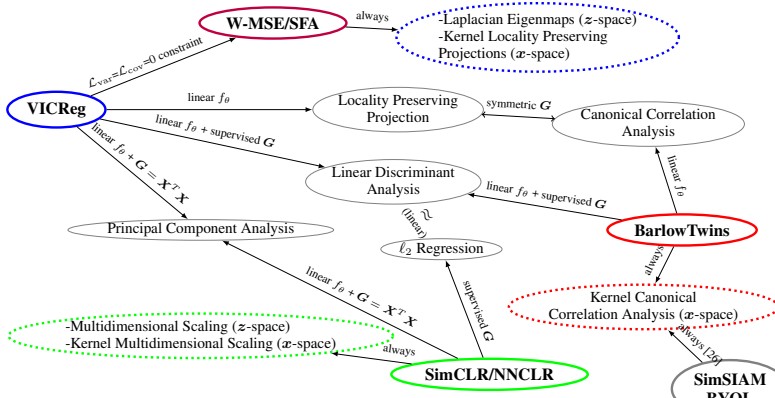

Figure 1: Summary of our unification of SSL methods to known *local* and *global* spectral embedding methods. In doing so, we are able to find the exact settings for which different methods provably become identical. In short, all are concerned in preserving the left-singular vectors of the similarity matrix $G$ (see Fig. 6) in the representation $Z$.

existing methods e.g. validating some empirically found tricks as necessary conditions for stable gradient dynamics [13–16], (ii) studying the role of each SSL component e.g. the projector and predictor networks [17–19], or (iii) developing novel SSL criteria that often combine multiple interpretable objectives that a SSL model must fulfill [20–25]. While those branches have led to novel understandings and even stem novel SSL methods, some fundamental questions remain open.

More recently, a few focused studies have started to provide theoretical works e.g. Tian [27] on contrastive losses with deep linear networks, HaoChen et al. [28, 29] on SimCLR [8], [30] on the projector of contrastive models, [14] on BYOL [31] and SimSIAM [8].Those studies paved our way forward as *we propose in this paper a much broader analysis that applies to most (if not all) existing SSL methods allowing us for the first time to provide provable design guildelines to practitioners in their choice of architecture and methods*. We do so by unifying most SSL methods as different flavors of spectral methods for embedding and clustering as summarized in Fig. 1. The instrumental results we obtain allow us to answer some long-standing questions both when employing a linear model and an infinite capacity one which we summarize as part of our contributions below:

1. **Closed-form optimal representation for SSL losses.** The Deep Network (DN)representation $Z$ of inputs $X$ learned by minimizing *any* SSL loss given a sample relation matrix $G$ is obtained in closed-form, shedding light to many spectral properties of those representations e.g. SSL only constrains the left singular vectors and singular values of $Z$ to align with the ones of $G$ (Sections 3.1, 4.1 and 5.1 for VICReg, SimCLR and BarlowTwins).

2. **Closed-form optimal network parameters for SSL losses with linear networks.** The linear representation $Z = XW + b$ parameters obtained by minimizing *any* SSL loss given a sample relation matrix $G$ are obtained in closed-form, providing insights into the type of input statistics that a network parameters focuses on to produce the optimal input mapping (Sections 3.2 and 5.2 for VICReg and BarlowTwins).

3. **Exact equivalence between SSL and spectral embedding methods.** SSL methods employ diverse criterion that can be tied to eponymous spectral analysis methods both in embedding space and in data space, and with a nonlinear or a linear DN e.g. Laplacian Eigenmaps (VICReg, Section 3.2), ISOMAP (SimCLR/NNCLR, Section 4.3), Canonical Correlation Analysis (BarlowTwins, Section 5.2) and when employing a linear network as Locality Preserving Projection (VICReg), Cannonical Correlation Analysis (BarlowTwins), and Linear Discriminant Analysis for both VICReg and BarlowTwins (summarized in Fig. 1).

We also relegate to Appendix C a study quantifying the relationship between the optimal representation of each SSL method and the downstream classification task performances e.g. when the correct data relation matrix is given (Appendix C.3) as those results, although not crucial to our contributions, follow directly from the above ties between SSL and spectral embedding. *We carefully prove each statement of this study in Appendix F*.

## 2 Notations and Background on Self-Supervised Learning

We provide in this section a brief reminder of the main Self-Supervised Learning (SSL) methods, their associated losses, and the common notations that we will rely on for the remaining of the study.

**Dataset, Embedding and Relation Matrix Notations.** Regardless of the loss and method employed, SSL relies on having access to a set of observations i.e. input samples $\boldsymbol{X} \triangleq [\boldsymbol{x}_1, \dots, \boldsymbol{x}_N]^T \in \mathbb{R}^{N \times D}$ and a known pairwise positive relation between those samples e.g. in the form of a *symmetric* matrix $\boldsymbol{G} \in (\mathbb{R}^+)^{N \times N}$ where $(\boldsymbol{G})_{i,j} > 0$ iff samples $\boldsymbol{x}_i$ and $\boldsymbol{x}_j$ are known to be semantically related, and with 0 in the diagonal. Commonly, one is only given a dataset $\boldsymbol{X}'$ and artificially constructs $\boldsymbol{X}$ and $\boldsymbol{G}$ from augmentations of $\boldsymbol{X}'$ e.g. rotated, noisy versions of the original samples and turning the corresponding entries of $\boldsymbol{G}$ to be positive for the samples that have been augmented form the same original sample. Lastly, $\boldsymbol{Z} \in \mathbb{R}^{N \times K}$ denotes the matrix of feature maps obtained from a model $f_\theta : \mathbb{R}^D \mapsto \mathbb{R}^K$ —commonly a Deep Network— as $\boldsymbol{Z} \triangleq [f_\theta(\boldsymbol{x}_1), \dots, f_\theta(\boldsymbol{x}_N)]^T$.

**VICReg.** With the above notations out of the way, we can remind the *VICReg* loss as defined in Bardes et al. [25] as a function of $\boldsymbol{X}$ and $\boldsymbol{G}$ in the following triplet loss

$$\mathcal{L}_{\text{vic}} = \alpha \sum_{k=1}^{K} \max\left(0, 1 - \sqrt{\text{Cov}(\boldsymbol{Z})_{k,k}}\right) + \beta \sum_{j \neq k} \text{Cov}(\boldsymbol{Z})_{k,j}^2 + \frac{\gamma}{N} \sum_{i=1}^{N} \sum_{j=1}^{N} (\boldsymbol{G})_{i,j} \|\boldsymbol{Z}_{i,.} - \boldsymbol{Z}_{j,.}\|_2^2. \quad (1)$$

We will often refer to each term in Eq. (1) as $\mathcal{L}_{\text{var}}$, $\mathcal{L}_{\text{cov}}$, and $\mathcal{L}_{\text{inv}}$ respectively.

**SimCLR.** The SimCLR loss [8] is slightly different and first produces an estimated relation matrix $\widehat{\boldsymbol{G}}(\boldsymbol{Z})$ generally using the cosine similarity (CoSim) via

$$(\widehat{\boldsymbol{G}}(\boldsymbol{Z}))_{i,j} = \frac{e^{\text{CoSim}(\boldsymbol{z}_i, \boldsymbol{z}_j)/\tau}}{\sum_{j=1, j \neq i}^{N} e^{\text{CoSim}(\boldsymbol{z}_i, \boldsymbol{z}_j)/\tau}}, \quad (2)$$

with $\tau > 0$ a temperature parameter. Then SimCLR encourages the elements of $\widehat{\boldsymbol{G}}(\boldsymbol{Z})$ and $\boldsymbol{G}$ to match. The most popular solution to achieve that is to leverage the infoNCE loss given by

$$\mathcal{L}_{\text{SimCLR}} = - \sum_{i=1}^{N} \sum_{h=1}^{N} (\boldsymbol{G})_{i,j} \log(\widehat{\boldsymbol{G}}(\boldsymbol{Z}))_{i,j}. \quad (3)$$

The only difference between SimCLR and its varients e.g. NNCLR [32] lies in defining $\boldsymbol{G}$.

**BarlowTwins.** Lastly, *BarlowTwins* [24] proposes yet a slightly different approach where $\boldsymbol{z}_i$ must be close to $\boldsymbol{z}_j$ if $\boldsymbol{G}_{i,j} > 0$. They do so with different flavors of losses and constraints to facilitate training. Hence, and for those models only, it is common to explicitly group $\boldsymbol{X}$ into two subsets $\boldsymbol{X}_{\text{left}}$ and $\boldsymbol{X}_{\text{right}}$ based on $\boldsymbol{G}$ so that $((\boldsymbol{X}_{\text{left}})_n, (\boldsymbol{X}_{\text{right}})_n), \forall n$ are all the positive pairs from $(\boldsymbol{X}, \boldsymbol{G})$. This does not lose any generality. In fact, suppose that we have 5 samples $a, b, c, d, e$, and that $\boldsymbol{G}$ says that $a, b, c$ are related to each other, and that $d, e$ are related to each other. Then, we can create the two data matrices as

$$\boldsymbol{X}_{\text{left}} = [a, a, b, b, c, c, d, e], \ \boldsymbol{X}_{\text{right}} = [b, c, a, c, a, b, e, d]. \quad (4)$$

Once the two (left/right) views are obtained, the corresponding embeddings $\boldsymbol{Z}_{\text{left}}, \boldsymbol{Z}_{\text{right}}$ can be computed and the BarlowTwins is then defined as

$$\mathcal{L}_{\text{BT}} = \sum_{k=1}^{K} (\text{CoSim}((\boldsymbol{Z}_{\text{left}})_{.,k}, (\boldsymbol{Z}_{\text{right}})_{.,k}) - 1)^2 + \alpha \sum_{k=1, k' \neq k}^{K} \text{CoSim}((\boldsymbol{Z}_{\text{left}})_{.,k}, (\boldsymbol{Z}_{\text{right}})_{.,k'})^2. \quad (5)$$

where one should notice that those terms correspond to the cross-correlation matrix between the two embeddings.

Our goal in the following sections (Section 3 for VICReg, Section 4 for SimCLR/NNCLR, and Section 5 for BarlowTwins) will be to find the optimal representations $\boldsymbol{Z}$ of $\boldsymbol{X}$ in various regimes. Three surprising facts will emerge: (i) all existing methods recover exactly some flavors of famous spectral methods, (ii) the spectral properties of $\boldsymbol{Z}$ can be provably characterized for each methods, and (iii) from those properties, we will provide necessary and sufficient conditions for a SSL representation to perfectly solve a downstream task. We provide linear algebra notations in Appendix B that might be useful for readers unfamiliar with methods such as the Singular Value Decomposition (SVD) [33].

## 3 VICReg Minimizes the Dirichlet Energy to Produce Smooth Signals on the Graph $G$ While Preventing Dimensional Collapse

Recall from Section 2 and Eq. (1) that VICReg is defined as a triplet loss (variance/invariance/covariance). We first demonstrate in Section 3.1 that the optimal VICReg representation can be obtained in

closed form (Theorem 1) and that turning the VICReg optimization as a constrained problem recovers Laplacian Eigenmaps in embedding space and Kernel Locality Preserving Projection in data space (Section 3.2). We also consider the linear network regime where we obtain the analytical form of the optimal network's parameters (Theorem 3).

## 3.1 Close-Form Optimal Representation for VICReg

First, we build up some insights into VICReg by demonstrating how the invariance term corresponds to the Dirichlet energy of the signal $\boldsymbol{Z}$ on the graph $\boldsymbol{G}$. Then, replacing the variance hinge loss at 1 with the squared loss at 1 as in $\sum_{k=1}^{K} \left(1 - \text{Cov}(\boldsymbol{Z})_{k,k}\right)^2$, notice that minimizing the latter implies minimizing the former. With that, we are able to obtain the close-form optimal representation $\boldsymbol{Z}^*$ minimizing Eq. (1) only as a function of $\boldsymbol{G}$ and the loss' hyperparameters.

**From invariance to trace minimization.** The first insight that we propose into VICReg is obtained by rewriting the invariance loss of VICReg as the energy of the signal $\boldsymbol{Z}$ on the graph $\boldsymbol{G}$ [34] since we have (derivations in Appendix F.5)

$$\sum_{i=1}^{N} \sum_{j=1}^{N} (\boldsymbol{G})_{i,j} \|(\boldsymbol{Z})_{i,.} - (\boldsymbol{Z})_{j,.}\|_2^2 = 2 \text{Tr}\left(\boldsymbol{Z}^T \boldsymbol{L} \boldsymbol{Z}\right) \quad (\triangleq \text{Dirichlet energy of } \boldsymbol{Z} \text{ on } \boldsymbol{G}), \quad (6)$$

where $\boldsymbol{L}$ is the graph Laplacian matrix $\boldsymbol{L} = \boldsymbol{D} - \boldsymbol{G}$ with $\boldsymbol{D}$ the diagonal degree matrix of $\boldsymbol{G}$ i.e. $(\boldsymbol{D})_{i,j} = \sum_j (\boldsymbol{G})_{i,j}$ and $(\boldsymbol{D})_{i,j} = 0, \forall i \neq j$. From Eq. (6) it is clear that the invariance term depends on the matching between the left singular vectors of $\boldsymbol{Z}$ and the eigenvectors of $\boldsymbol{L}$. Hence, *non-contrastive learning aims at producing non-degenerate signals $\boldsymbol{Z}$ that are smooth on $\boldsymbol{G}$*.

**Optimal representation.** To gain further insights into VICReg, we ought to obtain the analytical form of the optimal representation $\boldsymbol{Z}^*$ minimizing Eq. (1) —although this optimum is not unique e.g. adding a constant entry to each column of $\boldsymbol{Z}$ does not change the loss value (details in Appendix F.6). We can now obtain the following characterization of $\boldsymbol{Z}^*$ as a function of the spectral decomposition of the matrix that combines two Laplacian matrices (details in Appendix F.7). The first, (left of Eq. (7)) comes form the variance+covariance term is the Laplacian of a complete graph i.e. where each node/sample is connected all others. The second comes from the SSL graph $\boldsymbol{G}$ to form the following (with its eigen-decomposition)

$$\underbrace{\boldsymbol{I} - \boldsymbol{1}\boldsymbol{1}^T/N}_{\text{Laplacian of a complete graph}} - \frac{\gamma}{\alpha} \underbrace{(\boldsymbol{D} - \boldsymbol{G})}_{\text{Laplacian of the SSL/sup. graph}} = \boldsymbol{P}_{\alpha,\gamma} \text{diag}(\boldsymbol{\lambda}_{\alpha,\gamma}) \boldsymbol{P}_{\alpha,\gamma}^T, \quad (7)$$

where the eigenvalues/eigenvectors are in descending orders. The eigenvectors of the combined Laplacians will be key to produce the optimal VICReg representation as formalized below.

**Theorem 1.** *A global minimizer of the VICReg loss ($\alpha = \beta, \forall \gamma$) denoted by $\boldsymbol{Z}_{\alpha,\gamma}^*$ is obtained from Eq. (7) along with the minimal achievable loss which are given by*

$$\boldsymbol{Z}_{\alpha,\gamma}^* = (\boldsymbol{P}_{\alpha,\gamma}(\text{diag}(\boldsymbol{\lambda}_{\alpha,\gamma})N)^{1/2})_{:,1:K} \quad \text{and} \quad \min_{\boldsymbol{Z} \in \mathbb{R}^{N \times K}} \mathcal{L}_{\text{VIC}} = \alpha(K - \|(\boldsymbol{\lambda}_{\alpha,\beta})_{1:K}\|_2^2),$$

*and any $K$-out-of-$N$ columns of $\boldsymbol{P}_{\alpha,\gamma}(\boldsymbol{\Lambda}_{\alpha,\gamma}N)^{1/2}$ is a local minimum. (Proof in Appendix F.7.)*

The above result provides a few key insights. First, only the ratio $\gamma/\alpha$ governs the VICReg representation. Second, there exists many local minimum, some of which can be explicitly found by taking various $K$-out-of-$N$ columns of $\boldsymbol{P}_{\alpha,\gamma}(\boldsymbol{\Lambda}_{\alpha,\gamma}N)^{1/2}$ which we display in Fig. 2 along with the loss landscape of $\mathcal{L}_{\text{VIC}}$ around the optimal representation $\boldsymbol{Z}^*$. We also depict in Fig. 3 the evolution of the eigenvalues $(\boldsymbol{\lambda}_{\alpha,\gamma})_{1:K}$ for varying $\gamma$ along with the downstream task (induced by $\boldsymbol{G}$) training performance. We observe that VICReg benefits from a sweet-spot where it can both preserve a full-rank representation $\boldsymbol{Z}^*$ and incorporate enough information about $\boldsymbol{G}$ to solve the task at hand perfectly.

Before moving to other SSL methods, we first emphasize the ability of VICReg to recover *local* spectral methods in the following sections.

## 3.2 VICReg Recovers Local Spectral Embedding Methods

The goal of this section is to relate VICReg to known *local* spectral embedding methods both in feature space ($\boldsymbol{Z}$) and in data space ($\boldsymbol{X}$).

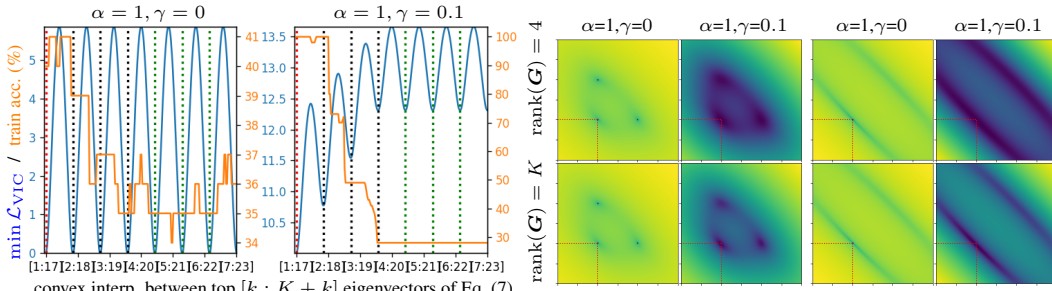

Figure 2: **Left:** depiction of the optimal VICReg loss with varying hyper-parameters (**blue line**) when the representation is formed from the top $[k : K + k - 1]$ eigenvectors of Eq. (7) with convex interpolation in-between. Recall from Theorem 1 that the global optimum is given by the $[1 : K - 1]$ case. We also depict the downstream task performance (**orange line**) and we clearly observe that both are closely related as expected (see Theorem 9). Notice that since we are considering classification, even without the correct first eigenvector the linear classifier on top of $\boldsymbol{Z}^*_{\alpha,\gamma}$ is able to solve the task at hand thanks to the probability constraint that must sum to 1 i.e. the last component can be recovered from the first $C - 1$. **Right:** depiction of the loss landscape of $\mathcal{L}_{\text{VIC}}$ around the optimal $\boldsymbol{Z}^*_{\alpha,\gamma}$ on the left using the directions provides by the top $[2 : K]$ and $[3 : K + 1]$ eigenvectors of Eq. (7), and then with random directions in $\boldsymbol{Z}$-space. All experiments employed $N = 256, K = 16, \text{rank}(\boldsymbol{G}) = 4$.

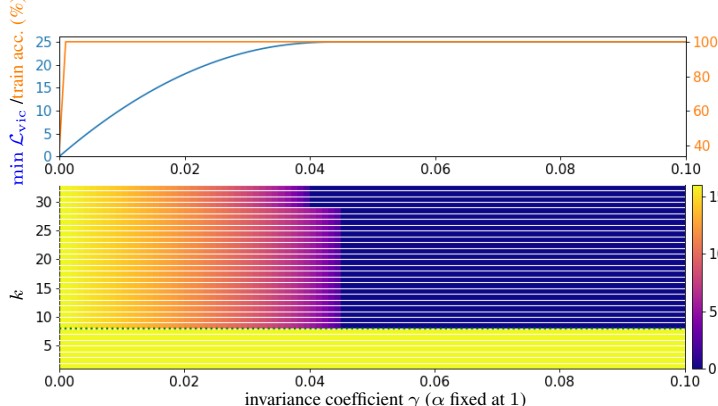

Figure 3: **Top:** minimal VICReg loss (**blue**) from Theorem 1 and corresponding downstream task performance (**orange**). **Bottom:** evolution of $\boldsymbol{Z}^*_{\alpha,\gamma}$'s singular values. *VICReg benefits from a $(\alpha, \gamma)$-zone for which $\boldsymbol{Z}^*_{\alpha,\gamma}$ remains full rank and incorporates enough information on $\boldsymbol{G}$ to solve the downstream task.* Hence, VICReg hyper-parameters $\gamma/\alpha$ should be adapted depending on the confidence one has into $\boldsymbol{G}$. All experiments employed $N = 256, K = 32, \text{rank}(\boldsymbol{G}) = 8$.

**In feature space.** Laplacian Eigenmaps (LE) [35] is a non-parametric method searching for a representation $\boldsymbol{Z}$ by minimizing the following Brockett [36] optimization problem

$$\min_{\theta:\boldsymbol{Z}^T\boldsymbol{D}\boldsymbol{Z}=\boldsymbol{I}} \text{Tr}\left(\boldsymbol{Z}^T\left(\boldsymbol{D} - \boldsymbol{G}\right)\boldsymbol{Z}\right), \tag{8}$$

with $\boldsymbol{D}$ the diagonal degree matrix of $\boldsymbol{G}$ (recall Eq. (7)).

**Theorem 2.** *Given a dataset $\boldsymbol{X}$ and relation matrix $\boldsymbol{G}$ solving the LE optimization problem Eq. (8) produces a representation that minimizes the VICReg loss with constraint that the variance and covariance loss are 0 as in $\mathcal{L}_{\text{vic}}(\boldsymbol{Z}^*_{\text{LE}})=\min_{\boldsymbol{Z}} \mathcal{L}_{\text{inv}}(\boldsymbol{Z})$ s.t. $\mathcal{L}_{\text{var}}=0$ and $\mathcal{L}_{\text{cov}}=0$. (Proof in Appendix F.8.)*

One important observation is that 8 recovers yet another SSL method known as W-MSE [37] which can now be seen as the constrained counterpart of VICReg, and in the linear regime, recovers Slow Feature Analysis (SFA) although without the dimension ordering, and is thus also closely related to its extension presented in Pfau et al. [38]. Furthermore, Eq. (8) and variants have been studied in Agrawal et al. [39] in the context of kernel PCA, some of which could provide interesting variations of the constrained VICReg setting. We also ought to highlight however that a crucial part of LE lies in the design of that matrix $\boldsymbol{G}$, often found from a $k$-NN graph [40] of the samples $\boldsymbol{X}$ in the input space, while in SSL it is constructed from data-augmentations, or given.

**In data space with DNs.** The difficulty to produce new representations $\boldsymbol{z}$ for new data samples (a shared difficulty among non-parametric methods) led to the development of a two-step modeling process [41] as $\boldsymbol{x} \, (\in \mathbb{R}^D) \mapsto \boldsymbol{h} = \phi(\boldsymbol{x}) \, (\in \mathbb{R}^S) \mapsto \boldsymbol{z} = \boldsymbol{W}^T\boldsymbol{h} \, (\in \mathbb{R}^K)$ with $S \gg K, \boldsymbol{W} \in \mathbb{R}^{M \times K}$,

and where $\phi$'s goal is to learn a generic input embedding that can be reused on new samples $\boldsymbol{x}$. To see this, we collect those mappings for all the training set into the matrix $\Phi \in \mathbb{R}^{N \times S}$. With that, the LE problem in data space —known as the kernel Locality Preserving Projection (kLPP) [42] problem— becomes

$$\min_{\theta: \boldsymbol{W}^T \Phi^T \boldsymbol{D} \Phi \boldsymbol{W} = \boldsymbol{I}} \operatorname{Tr}\left(\boldsymbol{W}^T \Phi^T \left(\boldsymbol{D} - \boldsymbol{G}\right) \Phi \boldsymbol{W}\right), \tag{9}$$

so that the original LE representation $\boldsymbol{Z}$ can be obtained simply as $\boldsymbol{Z} = \Phi \boldsymbol{W}$. And more importantly, given a new sample $\boldsymbol{x}$, one directly computes $\boldsymbol{z} = \boldsymbol{W}^T \phi(\boldsymbol{x})$. The crucial result of interest for our study is the following one that simply combines Theorem 2 with a result from He and Niyogi [42] demonstrating the equivalence between LE in feature space and KLLE in data space.

**Proposition 1.** *VICReg with variance/covariance constraint solves LE in embedding space and KLLE in input space (recall Eq. (9)) employing a DN for $\phi$.*

**In data space with linear models.** We now consider a linear mapping $\boldsymbol{Z} = \boldsymbol{XW}$ (the offset is taken care of by adding a $\boldsymbol{1}$ column to $\boldsymbol{X}$). In that case, VICReg with constrained $\mathcal{L}_{\mathrm{var}} = 0, \mathcal{L}_{\mathrm{cov}} = 0$ (as in Theorem 3)recovers two known spectral methods: Locality Preserving Projections (LPP) [42] for an arbitrary relation matrix $\boldsymbol{G}$, and Linear Discriminant Analysis (LDA) [43, 44] when $\boldsymbol{G}$ is the supervised relation matrix. In both cases we obtain the analytical form of the optimal weights $\boldsymbol{W}$, as long as the within class/cluster variance is positive.

**Theorem 3.** *Linear VICReg recovers LPP for any $\boldsymbol{G}$ and $K \leq N$, and recovers LDA for supervised $\boldsymbol{G}$ and $K = C$, in both cases the optimal parameter $\boldsymbol{W}^*$ is given by the top-$K$ eigenvectors of $(\boldsymbol{X}^T(\boldsymbol{D} - \boldsymbol{G})\boldsymbol{X})^{-1} \boldsymbol{X}^T \boldsymbol{G} \boldsymbol{X}$. (Proofs in Appendices F.13 and F.14.)*

Interestingly, the eigenvalues associated to the eigenvectors of $\boldsymbol{W}^*$ exactly recover the multivariate analysis of variance (MANOVA) sufficient statistics of the data [45, 46]. Hence, although not further explored in this study, we believe that important statistical results could be further obtained e.g. to assess the goodness-of-fit of the model without requiring a downstream task [47–49]. We now turn to another important SSL loss which is SimCLR and its variants.

# 4 SimCLR Solves a Generalized Multidimensional Scaling Problem à la ISOMAP

Recall from Section 2 and Eq. (2) that SimCLR first computes a similarity matrix $\widehat{\boldsymbol{G}}$ of some flavor that depends on the representation $\boldsymbol{Z}$, and then matches it against the known data relation $\boldsymbol{G}$. Different $\boldsymbol{Z} \mapsto \widehat{\boldsymbol{G}}$ methods lead to different variants of SimCLR [8] such as NNCLR [32] or MeanShift [50]. The goal of this section is two-fold. First, we demonstrate in Section 4.1 that different $\boldsymbol{Z} \mapsto \widehat{\boldsymbol{G}}$ mappings are solutions of different optimization problems (Theorem 4) —all trying to estimate the similarity matrix $\boldsymbol{G}$ from the signals $\boldsymbol{Z}$ akin to Laplacian estimation in Graph Signal Processing. Second, we demonstrate in Section 4.2 that SimCLR and its variants force $\boldsymbol{Z}$'s spectrum to align with the one of $\boldsymbol{G}$ as the training task falls back to a generalized (kernel) Multi-Dimensional Scaling method (Proposition 2).

## 4.1 Step 1: SimCLR Pairwise Similarities Solve a Graph Laplacian Estimation Problem

Let's first define the minimization problem that given a set of signals i.e. rows of $\boldsymbol{Z}$ produces a relation estimate $\widehat{\boldsymbol{G}}$ of $\boldsymbol{G}$. To ease notations, we gather in the $N \times N$ matrix $\boldsymbol{D}$ all the pairwise distances $(\boldsymbol{D})_{i,j} = d(f_\theta(\boldsymbol{x}_i), f_\theta(\boldsymbol{x}_j))$ with $d$ any preferred metric. The standard problem of estimating $\widehat{\boldsymbol{G}}$ from $\boldsymbol{Z}$ can be cast as an optimization problem [51, 52] as

$$\widehat{\boldsymbol{G}}_{d,\mathcal{R}} = \underset{\boldsymbol{G} \in \mathcal{G}}{\arg\min} \sum_{i,j} d(f_\theta(\boldsymbol{x}_i), f_\theta(\boldsymbol{x}_j))(\boldsymbol{G})_{i,j} + \mathcal{R}(\boldsymbol{G}) = \underset{\boldsymbol{G} \in \mathcal{G}}{\arg\min} \operatorname{Tr}(\boldsymbol{D}\boldsymbol{G}) + \mathcal{R}(\boldsymbol{G}), \tag{10}$$

with $\mathcal{G}$ the set (or subset) of symmetric matrices with nonnegative entries and zero diagonal, and with $\mathcal{R}$ a regularizer preventing $\widehat{\boldsymbol{G}}$ to be the trivial zero matrix e.g.

$$\mathcal{R}_{\log}(\boldsymbol{G}) = \sum_{i \neq j} \tau \boldsymbol{G}_{i,j}(\log(\boldsymbol{G}_{i,j}) - 1) \quad \text{or} \quad \mathcal{R}_{\mathrm{F}}(\boldsymbol{G}) = \sum_{i \neq j} \tau \boldsymbol{G}_{i,j}(\boldsymbol{G}_{i,j} - 1). \tag{11}$$

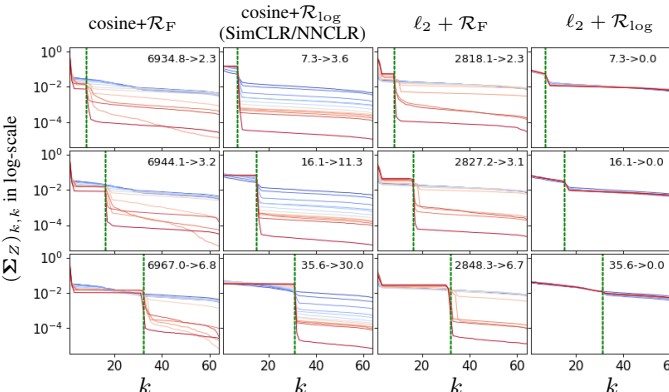

Figure 4: Depiction of the representation's singular values during training (**from blue to red** with the SimCLR loss with $\mathcal{G}_{\mathrm{rsto}}$, varying $\mathrm{rank}(\boldsymbol{G}) \in \{8, 16, 32\}$ (**rows, green dotted lines**), number in top-right corner) with various $(d, \mathcal{R})$ configurations (**columns**, recall Theorem 4). The rank of the learned representation matches exactly the one of $\boldsymbol{G}$ validating the result from Theorem 5 regardless of the chosen hyper-parameters.

We provide in Fig. 7 a depiction of the impact of $\mathcal{R}(\boldsymbol{G})$ which pushes the entries of the weight matrix to be close to 1 with strength depending on the temperature parameter $\tau$. Hence $\widehat{\boldsymbol{G}}$ from Eq. (10) is the optimal graph —expressed as a weight matrix— for which the signal $\boldsymbol{Z} = f_\theta(\boldsymbol{X})$ on that graph is smooth. For example one can solve Eq. (10) only on $\mathcal{G}_{\mathrm{rsto}}$, the space of right-stochastic matrices i.e. a subset of $\mathcal{G}$ that only contains matrices whose rows sum to 1 i.e. $\mathcal{G}_{\mathrm{rsto}} = \{\boldsymbol{G} \in \mathcal{G} : \boldsymbol{G}\mathbf{1} = \mathbf{1}\}$.

**Theorem 4.** *Using $\mathcal{R}$ from Eq. (11) leads to the following graph weight estimate*

$$(\widehat{\boldsymbol{G}}_{d,\mathcal{R}_{\log}})_{i,j} = e^{\frac{-1}{\tau}d(f_\theta(\boldsymbol{x}_i),f_\theta(\boldsymbol{x}_j))}\mathbf{1}_{\{1\neq j\}}, \qquad\qquad \textit{(with $\mathcal{G}$)}$$

$$(\widehat{\boldsymbol{G}}_{d,\mathcal{R}_{\log}})_{i,j} = \frac{e^{\frac{-1}{\tau}d(f_\theta(\boldsymbol{x}_i),f_\theta(\boldsymbol{x}_j))}}{\sum_{j\neq i}e^{\frac{-1}{\tau}d(f_\theta(\boldsymbol{x}_i),f_\theta(\boldsymbol{x}_j))}}\mathbf{1}_{\{1\neq j\}}, \qquad\qquad \textit{(with $\mathcal{G}_{\mathrm{rsto}}$)} \qquad (12)$$

*and thus if $d$ is the cosine distance, Eq. (12) recovers SimCLR's case. (Proof in Appendix F.9.)*

Based on this result, deriving novel, principled and interpretable variations of SimCLR is streamlined as we demonstrate in Appendix E by solving Eq. (10) with different constraints. Now that we understood the first part of the SimCLR method, we move to the second part, matching that graph estimate to the given one.

## 4.2  Step 2: SimCLR Fits the Estimated Graph $\widehat{\boldsymbol{G}}$ to the Known Graph $\boldsymbol{G}$

After SimCLR estimates the graph with $\widehat{\boldsymbol{G}}$ as per the previous section, it employs a loss to enforce $\widehat{\boldsymbol{G}}$ to be as close as possible to $\boldsymbol{G}$ with some desired metric. That metric should reflect the properties that $\boldsymbol{G}$ fulfills e.g. being a doubly-stochastic, right-stochastic or else. We demonstrate in this section that in doing so, SimCLR forces $\boldsymbol{Z}$ to have the same nonzero left singular vectors as the nonzero eigenvectors of $\boldsymbol{G}$, and that as opposed to VICReg, the rank of $\boldsymbol{Z}_\tau^*$ and $\boldsymbol{G}$ always matches.

Let's first denote the SimCLR contrastive loss to be one of the two following variants (depending on the type of constraints put on $\widehat{\boldsymbol{G}}$ and $\boldsymbol{G}$

$$\mathcal{L}_{\mathrm{SimCLR}} = \|\boldsymbol{G} - \widehat{\boldsymbol{G}}\|_F^2 \text{ or } -\frac{1}{N}\sum_{n=1}^{N}\sum_{n'=1}^{N}(\boldsymbol{G})_{n,n'}\log((\widehat{\boldsymbol{G}})_{n,n'}). \qquad (13)$$

When minimizing Eq. (13), SimCLR will learn an embedding $\boldsymbol{Z}$ so that the graph estimate matches closely the known graph $\boldsymbol{G}$. This already brings a contrast from VICReg showing that instead, *SimCLR learns to produce signals such that the graph estimate is close to the known graph.* We now characterize the optimal SimCLR representation $\boldsymbol{Z}_\tau^*$, recalling that we denote by $\boldsymbol{U}_G$ and $\boldsymbol{\Sigma}_G$ the left singular vectors and singular values of $\boldsymbol{G}$ respectively. Notice that since $\boldsymbol{G}$ is symmetric semi-definite positive, $\boldsymbol{U}$ also corresponds to its eigenvectors and $\boldsymbol{\Sigma}_G^2$ to its eigenvalues.

**Theorem 5.** *A global minimizer of the SimCLR loss denoted by $\boldsymbol{Z}^*$ along with the minimal achievable loss using Eq. (16) or Eq. (17), $\tau \geq \max_{i,j}(D)_{i,j}$ and with the LHS of Eq. (13) are given by*

$$\boldsymbol{Z}_\tau^* = (\boldsymbol{U}_G\boldsymbol{\Sigma}_G^{1/2})_{:,1:K} \quad \textit{and} \quad \min_{\boldsymbol{Z}\in\mathbb{R}^{N\times K}}\mathcal{L}_{\mathrm{SimCLR}} = \sum_{k=K+1}^{N}(\boldsymbol{\Sigma}_G^2)_{k,k},$$

*up to permutations of the singular vectors associated to the same singular value. Also, for any loss of Eq. (13) and graph estimation, the rank of $\boldsymbol{Z}_\tau^*$ is $\min(K, \mathrm{rank}(\boldsymbol{G}))$. (Proof in Appendix F.10.)*

We illustrate the above theorem for many combinations of distances and regularizers in Fig. 4 where we see that in all cases, SimCLR forces the representations $\boldsymbol{Z}$ to have a dimensional collapse, a phenomenon first observed in Hua et al. [17] and that has been one of the unanswered phenomenon in SSL [20, 30].

In our goal to unify SSL methods under the helm of spectral embedding methods, we now propose the following section that ties SimCLR and its variants to *global* spectral methods.

### 4.3 SimCLR Recovers Global Spectral Embedding Methods

We now propose to tie the SimCLR method along with its variants e.g. NNCLR to known *global* spectral methods, e.g. ISOMAP [53] which is in contrast to VICReg which was tied to *local* spectral methods (recall Section 3.2).

**In feature space.** Let's first recall that ISOMAP is a variation of Multi-Dimensional Scaling (MDS) [54] also known as Principal Coordinates Analysis. Classical MDS tries to learn embedding vectors that have similar pairwise distance (usually $\ell_2$) than the pairwise distance of the given input data. Often, MDS does this by using similarities instead of distances and thus by solving the following optimization problem $\min_{\boldsymbol{Z}} \|\boldsymbol{G} - \boldsymbol{Z}\boldsymbol{Z}^T\|_F^2$. At the most general level, ISOMAP simply corresponds to solving that same optimization problem but after redefining $\boldsymbol{G}$ to better capture the geometric information of $\boldsymbol{X}$ e.g. using the shortest path distance of the $k$-NN graph of $\boldsymbol{X}$ [55]. The surprising result that we formalize below is that SimCLR and its variants recover ISOMAP.

**Proposition 2.** *SimCLR, using the settings of Theorem 5, recovers ISOMAP (and MDS for the correct choice of $\boldsymbol{G}$). (Proof in Appendix F.11.)*

**In data space with DNs.** From the above, we can extend Proposition 2 but in input space, in a very similar way as was done in Section 3.2. In fact, originating in Webb [56], there was a search to extend MDS, and ISOMAP to an input space formulation to solve the out-of-bag problem. In this setting, and taking MDS as an example, the original similarity matrix $\boldsymbol{Z}\boldsymbol{Z}^T$ is replaced with $\Phi\boldsymbol{W}^T\boldsymbol{W}\Phi^T$ using the same notations as in Eq. (9) and already known relationship between those models, we obtain the following.

**Proposition 3** ([57]). *Whenever SimCLR recovers ISOMAP or MDS in feature space, it recovers kernel ISOMAP or kernel PCA [58] in input space.*

We now ought to turn to BarlowTwins, another non-contrastive method akin to VICReg (both of which fall back to LDA in the linear regime and with supervised $\boldsymbol{G}$).

## 5 BarlowTwins Solves a (Kernel) Canonical Correlation Analysis Problem and Can Recover VICReg

Our last step in our journey to unify SSL methods under spectral embedding methods deals with BarlowTwins. Akin to the development for VICReg and SimCLR, BarlowTwins will also fall back to a known spectral method in embedding space (Section 5.1) and in data space (Section 5.2) where in the later case we again obtain the close-form optimal network parameters in the linear regime.

### 5.1 BarlowTwins Recovers Kernel Canonical Correlation Analysis

Recall from Section 2 and Eq. (5) that the BarlowTwins loss is based on a cross-correlation matrix between positive pairs of samples. As we did for VICReg and SimCLR, our goal here is to tie BarlowTwins to a known spectral method known as Kernel Canonical Correlation Analysis.

Although we focus here on BarlowTwins for clarity. We hope that our results on BarlowTwins will stem the unification of those methods too in future works. Going back to BarlowTwins, we now obtain the following result that nicely parallels with the ones we obtained for VICReg and SimCLR. In data space, BarlowTwins can be regarded (put in perspective with Section 5.2) as a nonlinear canonical correlation analysis (NLCA) [59] and in particular Kernel CCA (KCCA) [60, 61] akin to

how VICReg recovered Kernel Locality Preserving Projection and SimCLR Kernel ISOMAP. We leverage the same notations as in Section 3.2.

**Theorem 6.** *BarlowTwins recovers Kernel Canonical Correlation Analysis with a DN as the featurizer $\phi$ and produce a representation with rank $\min(K, D)$ for any value of $\alpha$ (recall Eq. (5)) and with orthogonal columns. (Proof in Appendix F.12.).*

We thus obtain from the above that BarlowTwins employing a DN featurized is akin to the Deep CCA [62] that proposed this setting exactly, and further akin to SimSIAM and BYOL [31] since Lee et al. [26] related the latter to Deep CCA. In addition to those links, the above provides further interpretation into the BarlowTwins' loss e.g. the additional $\epsilon$ constant added in the denominator of the BarlowTwins loss further corresponds to a ridge-type regularization which as been introduced in Gretton et al. [63] as a mean to introduce numerical stability.

The above statement also brings yet another flavor of SSL methods. In fact, where VICReg allows to control the rank of $\boldsymbol{Z}$ to be in-between $K$ and $\mathrm{rank}(\boldsymbol{G})$ throug the loss hyper-parameters, and where SimCLR enforces the rank of $\boldsymbol{Z}$ to be exactly the rank of $\boldsymbol{G}$, BarlowTwins enforces to have a full-rank representation. We depict in Fig. 8 the evolution of $\mathrm{rank}(\boldsymbol{Z})$ depending on the rank of the initialized representation $\mathrm{rank}(\boldsymbol{Z}_{\mathrm{init}})$ using a gradient descent optimizer. We see that if the initialization is full rank but $\boldsymbol{G}$ is lower rank, the BarlowTwins loss does not collapse the extra nonzero singular values of $\boldsymbol{Z}$. Vice-versa, if $\mathrm{rank}(\boldsymbol{Z}_{\mathrm{init}}) < \mathrm{rank}(\boldsymbol{G})$ then BarlowTwins loss will increase the rank of $\boldsymbol{Z}$. lastly, although not further studied here, we should point out to the reader that regularized forms of KCCA can be shown to include kernel ridge regression and regularized kernel Fisher LDA as special cases [64], further tying the special cases for which different SSL methods would fall back to the same model.

In the following section we will demonstrate how BarlowTwins in the linear regime exactly recovers Canonical Correlation Analysis.

## 5.2 With a Linear Network BarlowTwins Recovers Canonical Correlation Analysis and Linear Discriminant Analysis

The goal of this section is to further demonstrate the benefits of connecting SSL methods to spectral methods by exploiting the known techniques of the latter to help answer questions on the former.

As was done VICReg (we use linear settings of Section 3.2), we now obtain the optimal weights for BarlowTwins in the linear regime. We can even provide additional insights in this case since BarlowTwins is often seen as a key method that allows the use of different parameters/architectures to process $\boldsymbol{X}_{\mathrm{left}}$ and $\boldsymbol{X}_{\mathrm{right}}$. We now show under what conditions on $\boldsymbol{G}$ sharing parameters is sufficient by first demonstrating how BarlowTwins recovers exactly CCA, and even LDA for supervised $\boldsymbol{G}$. To streamline notations, we assume that our data is already centered, and thus define the covariance and cross-covariance matrices as $\boldsymbol{C}_{ll} = \boldsymbol{X}_{\mathrm{left}}^T \boldsymbol{X}_{\mathrm{right}}, \boldsymbol{C}_{lr} = \boldsymbol{X}_{\mathrm{left}}^T \boldsymbol{X}_{\mathrm{right}}$ and so on.

**Theorem 7.** *In the linear regime BarlowTwins recovers CCA with optimal weights given by*

$$\boldsymbol{W}_{\mathrm{left}}^* = \text{top-}K \text{ eigenvectors of } \boldsymbol{C}_{\mathrm{ll}}^{-1}\boldsymbol{C}_{\mathrm{lr}}\boldsymbol{C}_{\mathrm{rr}}^{-1}\boldsymbol{C}_{\mathrm{rl}} \text{ and } \boldsymbol{W}_{\mathrm{right}}^* = \boldsymbol{C}_{\mathrm{rr}}^{-1}\boldsymbol{C}_{\mathrm{rl}}\boldsymbol{W}_{\mathrm{left}}^*,$$

*and (i) —with a symmetric $\boldsymbol{G}$, weight-sharing naturally occurs— as the optimal weights are $\boldsymbol{W}_{\mathrm{left}}^* = \boldsymbol{W}_{\mathrm{right}}^* = \text{top-}K$ eigenvectors of $\boldsymbol{C}_{\mathrm{rr}}^{-1}\boldsymbol{C}_{\mathrm{rl}}$ and (ii) if $\boldsymbol{G}$ is supervised and $K = C$ then BarlowTwins recovers LDA and thus constrained VICReg (recall Theorem 3). (Proof in Appendix F.15.)*

The above result opens new venues to extend current SSL methods (BarlowTwins in this case). For example, penalized matrix decomposition (PMD) from Witten et al. [65] formulates a novel sparse formulation of CCA. In our context, this could lead to a new variation of BarlowTwins, in both the linear and nonlinear regimes. With the above results, we now connected most SSL methods to spectral methods, and found key properties that their representations/parameters inherit. We provide for completeness a few direct results in Appendix C that directly leverage the above results to characterize downstream performances of each method.

## 6 Conclusions and Limitations

We provided a unifying analysis of the major self-supervised learning methods covering VICReg (Section 3), SimCLR (Section 4) and BarlowTwins (Section 5) along with SimSIAM and BYOL

thanks to already existing results tying those to Deep CCA. In doing so, we were able to find the commonalities between all those methods and to provide general guidelines on how to derive alternative SSL methods from first principles. At a more general level, we were able to parallel the many SSL methods to global and local methods in spectral methods respectively. Among the many insights that we obtained, the most crucial one is that VICReg enables a continue control on collapsing the representation's rank versus encapsulating information about $G$. This is in contrast to BarlowTwins and SimCLR that either always maintain full-rank, or always collapse the representation none of which would be ideal. One major limitation is that for the nonlinear regime, we study the case of an infinite capacity model i.e. no implicit bias coming from the architecture comes into play. A potentially insightful future work would thus be to perform a similar analysis as the one provided here but including the implicit bias on the nonlinear mapping that different architecture exhibit. This way, we would obtain not only insights into the various SSL methods but also in their combination with various model architectures.

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
