# Appendix:
# Contrastive and Non-Contrastive Self-Supervised Learning Recover Global and Local Spectral Embedding Methods

1. **W-MSE** enforces unit variance and identity covariance of the representation while minimizing an invariance term for the positive pairs in $G$. The variance term removes the scaling ambiguity akin to PCA by setting the representation variances (per-dimension) to 1 while the covariance term enforces orthogonal representations to maximize the information content of the representation (provable in the linear regime). W-MSE recovers LE in the infinite model capacity regime (Theorem 2), and LDA and LPP in the linear model regime (Theorem 3). In this setting, the representation is always full-rank and thus should be preferred when the correct graph $G$ is estimated.

    - In the linear regime, this recovers eponymous methods such as Slow-Feature Analysis [66, 38] especially in the video setting where consecutive frames are considered as positive pairs, and can also be seen as variants of robust PCA [67] with known relationships between the corrupted signals that should have the same representation
    - In the linear regime with $0$ weighting of the invariance term, this recovers exactly PCA.

2. **BarlowTwins** proposes an alternative yet equivalent formulation in term of CCA (Theorem 6). In fact, the optimal representations of W-MSE and BarlowTwins are identical (Theorem 7). Hence, the representation is also always full-rank and thus should be preferred when the correct graph $G$ is estimated. **BYOL, SimSIAM** can be seen as variations of the CCA model of BarlowTwins as per Lee et al. [26].

3. **VICReg** removes the variance and covariance hard-constraints of W-MSE/BarlowTwins to turn them into regularizers. Hence, the optimal representation of Eq. (10) is no longer always whitened depending on the weighting of the three terms. In this setting, the representation has a rank that varies with the loss coefficients which should be adapted based on the confidence in the graph $G$.

4. **SimCLR** takes a different approach and propose instead to (i) assume that the produced representation lives on a graph on which it is smooth, (ii) estimate that graph from the representation, and (iii) ensure that this estimate is as close as possible to the observed graph $G$

5. The 1., 2., and 3. methods learn a representation by minimizing the Dirichlet energy of the representation on the graph $G$ while preventing collapse, while the method 4. acts differently. It first estimates $G$ from the representation using a graph Laplacian estimate with smoothness regularization, and then ensures that this estimate matches $G$.

The supplementary materials is providing the proofs of the main's paper formal results, and additional content e.g. Appendix C focuses on using our results to study downstream task performance of SSL methods. We also provide as much background results and references as possible throughout to ensure that all the derivations are self-contained. Some of the below derivation do not belong to formal statements but are included to help the curious readers get additional insights into current SSL methods.

## A   Experimental Details

For the evolution of the spectrum and the VICReg loss landscape (Figs. 2 and 3), we employed $N = 256$ data samples, $C = 8$ classes (i.e. connected components in $G$ and $K = 32$ representation dimensions. The evolution of the $\gamma$ parameter goes from 0.0 to 0.1 in 100 steps. For the dimensional collapse experiments (Figs. 4 and 8), we take $N = 512$ data samples and $K = 64$ dimensions for the representation and varying $G$ rank for each column of the corresponding figures.

## B   Linear Algebra Notations

This study heavily relies on the Singular Value Decomposition (SVD) of matrices [33, 68] e.g. $\boldsymbol{X} = \boldsymbol{U}_x \boldsymbol{\Sigma}_x \boldsymbol{V}_x^T$ that are denoted as the left singular vectors $\boldsymbol{U}_x \in \mathbb{R}^{N \times N}$, the singular values $\boldsymbol{\Sigma}_x \in \mathbb{R}^{N \times D}$ and the right singular

vectors $V_x \in \mathbb{R}^{D \times D}$ of $X \in \mathbb{R}^{N \times D}$. We will always specify as a lower-script and lower-case the matrix that is being decomposed ($x$ in this case). It will also be convenient to only consider the left/right singular vectors whose associated singular values are 0 that we will denote as $\overline{U}_x$ and $\overline{V}_x$ respectively. Conversely, the left/right singular vectors whose associated singular values are $> 0$ will be denote as $\widehat{U}_x$ and $\widehat{V}_x$ respectively. Lastly, we will denote by $\boldsymbol{\sigma}_z$ the vector of singular values such as $\boldsymbol{\Sigma}_z = \mathrm{diag}(\boldsymbol{\sigma})$ and without loss of generality and unless otherwise stated, this will always be in descending order.

## C Optimality of Self-Supervised Methods to Solve Downstream Tasks

The goal of this section is to answer the following question: *given a task —encoded as a target matrix—* $Y \in \mathbb{R}^{N \times C}$, *what are the sufficient statistics of* $Y$ *that a representation* $Z \in \mathbb{R}^{N \times K}$ *must preserve to ensure that* $\min_{W,b} \|Y - ZW - b\|_F^2 = 0$. The first Appendix C.1 will derive the optimal linear parameters (possibly non-unique) that minimize that loss, and will highlight the spectral properties of $Y$ that must be consistent in $Z$. From that, Appendix C.2 will be able to provide necessary and sufficient conditions for $Z$ to be optimal —in term of its left-singular vectors— and finally, Appendix C.3 extends the recent studies of Bao et al. [69], HaoChen et al. [28, 29] to multiple SSL methods by demonstrating how and when SSL representations are optimal for downstream tasks.

### C.1 Characterizing a Representation Usefulness by its Ability to Linearly Solve a Task

The goal of this section is to characterize the (possibly non-unique) parameter $W$ of the linear transformation that given any target matrix $Y$ and representation $Z$ minimize the Mean-Squared Error (MSE). Results in this section are standard in the linear algebra literature e.g. see Golub and Reinsch [70] but we include it for completeness. We should also highlight that although for classification the cross-entropy loss is more common, it has recently been showed that $\ell_2$ could perform as well [71] with the correct parameter tuning, even on large models e.g. on the Imagenet dataset with modern deep learning architectures. Hence, we will leverage that loss function throughout this section.

Hence we consider that we are given representations $z \triangleq f(x)$ from an input sample $x$. Hence, we will denote the dataset representation as $Z = [f(x_1), \ldots, f(x_N)]^T \in \mathbb{R}^{N \times K}$ given a dataset $\mathbb{X} \triangleq \{x_1, \ldots, x_N\}, N \in \mathbb{N}^*$. We are also given a task —encoded as a target matrix— $Y \triangleq [y_1, \ldots, y_N]^T \in \mathbb{R}^{N \times C}$ where each $y_n$ is associated to each input $x_n$.

Without loss of generality and to lighten our derivations, we will omit the bias vector $b$, it can be learned as part of $W$ by adding 1 to the features of $z_n$. Our loss function thus takes the following form $\mathcal{L}(W) = \frac{1}{2}\|Y - ZW\|_F^2$. To minimize this loss function, we will first find what matrices $W$ make the gradient of $\mathcal{L}$ with respect to $W$ vanish. Hence, we need to find $W \in \mathbb{R}^{K \times C}$ such that

$$\nabla_W \mathcal{L} = 0 \iff -Z^T(Y - ZW) = 0 \iff Z^T Y = Z^T Z W.$$

If we assumed $Z$ to be full rank, we would directly recover the usual least square solution $W^* = (Z^T Z)^{-1} Z^T Y$. However, we would like to (i) avoid any assumption on the spectrum of $Z$, and (ii) avoid the use of any standard regularizer such as Tikhonov that is commonly used to recover a unique solution to an ill-posed optimization problem. In fact, our goal is to find the (possibly non-unique) family of parameters $W^*$ that fulfill $\nabla_W \mathcal{L} = 0$ regardless of the properties of $Z$ and $Y$. To that end, we first start by using the SVD of $Z = U_z \Sigma_z V_z^\top$ —which always exists— to reformulate the above equality (see derivations in Appendix F.2) into

$$Z^T Y = Z^T Z W \iff W \in \left\{ (V_z \Sigma_z^{-1} U_z^T + \overline{V}_z M) Y : M \in \mathbb{R}^{K \times C} \right\}, \tag{14}$$

with $\overline{V}_z$ the $K \times (K - \mathrm{rank}(Z))$ matrix that horizontally stacks the right singular vectors of $Z$ which have their corresponding singular value 0, with the special case that $M = 0 \iff \mathrm{rank}(Z) = K$. We also slightly abuse notations and define $\Sigma_z^{-1}$ to be the $K \times N$ matrix which is zero for all off-diagonal elements, and with

$$(\Sigma_z^{-1})_{i,j} \triangleq \begin{cases} \frac{1}{(\sigma_z)_i} \iff i = j \wedge (\sigma_z)_i > 0 \\ 0 \text{ otherwise} \end{cases},$$

hence $\Sigma_z^{-1} \Sigma_z$ is a $K \times K$ matrix which is identity iff $X$ is full rank, and is otherwise filled with $K - \mathrm{rank}(X)$ zeros and $\mathrm{rank}(X)$ ones in the diagonal. On the other hand, $\Sigma_z \Sigma_z^{-1}$ is a diagonal $N \times N$ matrix with $N - \mathrm{rank}(X)$ zeros and $\mathrm{rank}(X)$ ones in the diagonal. Note that in the special case where $X$ is full-rank, $M = 0$ and $V_z \Sigma_z^{-1} U_z^T = (Z^T Z)^{-1} Z^T$ (see derivations in Appendix F.2) recovering the standard least-square solution. In fact, by plugging back in $W^*$ in the loss it is direct to obtain the equality (see derivations in Appendix F.3) $V_z \Sigma_z^T U_z^T Y = V_z \Sigma_z^T \Sigma_z V_z^\top W^*$.

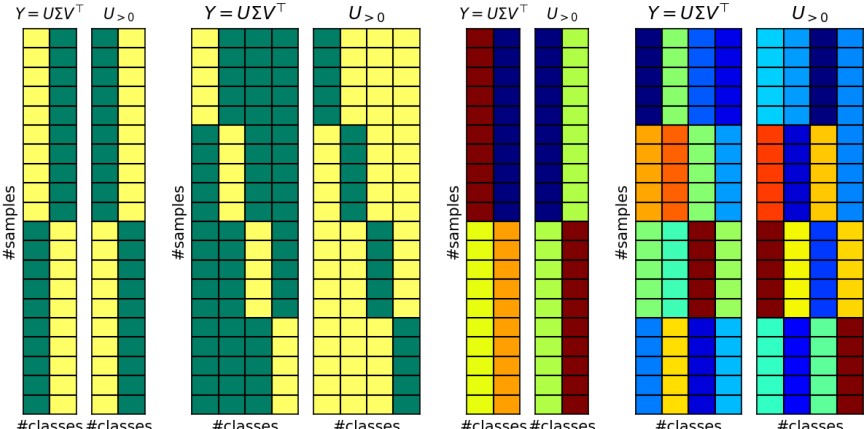

Figure 5: Depiction of typical target matrices $Y$ and their corresponding left singular vectors. Note that in this case all the nonzero singular values are identical, but with class-imbalance the singular values would be proportional to the class proportions. Hence it is clear that approximating the left singular vectors of $Y$ via SSL training (recall Theorem 9) the learned representations will exhibit per-class clustering —as long as the left-singular vectors of $G$ correctly encode the task.

## C.2   Necessary and Sufficient Conditions for Optimality of a Representation

Given a target matrix $Y$ and its SVD $U_y \Sigma_y V_y^T$, we obtain the following formal statement that demonstrates how the left-singular vectors of $Y$ must related to the left-singular vectors of $Z$ to allow for $\min_W \mathcal{L}(W)$ to be 0. This statement plays a crucial role in our study as it answers the question *what property $Z$ must fulfill —regardless on how it was produced— to guarantee that $0$ training error is achievable on the considered task?*

**Theorem 8** (Necessary and sufficient condition). *Given a task $Y \in \mathbb{R}^{N \times C}$ and a representation $Z \in \mathbb{R}^{N \times K}$ —with left-singular vectors associated to nonzero singular values denoted as $\widehat{U}_z$— the minimum linear loss is given by*

$$\min_{W \in \mathbb{R}^{K \times C}} \mathcal{L}(W) = \frac{1}{2}\|Y\|_F^2 - \frac{1}{2}\|\widehat{U}_z^T U_y \Sigma_y\|_F^2, \tag{15}$$

*which is $0$ iff the columns of $\widehat{U}_z$ spans the columns of $\widehat{U}_y$. (Proof in Appendix F.4.)*

The proof consists in using the solution $W$ from the set in Eq. (14), and after a few algebraic manipulations, Theorem 9 result is obtained. From that result alone we already obtain an interesting requirement on $Z$, namely that its top-$K$ left singular vectors must be the same —up to a rotation— to the top-$K$ left singular of $Y$. Theorem 9 also provides us with a direct necessary condition but not sufficient condition for optimality.

Understanding the inter-play between $\text{rank}(Z)$, $U_z$ and $U_y$ will play a crucial role in the next section where we propose to study self-supervised learning criterion, and their ability to produce optimal representations.

## C.3   Contrastive and Non-Contrastive Learning can all be Optimal

We now demonstrate in this section that any representation $Z$ learned by any of the SSL method (VICReg, BarlowTwins, SimCLR) can be optimal for a downstream task, as long as the data geometry encoded in $G$ follows the left-singular vectors of $Y$, the target matrix which embodies the considered downstream task.

**Theorem 9.** *Given a dataset $X$ and relation matrix $G$, minimizing the VICReg -or- SimCLR -or- BarlowTwins loss produces a representation that is optimal for a task $Y = U_y \Sigma_y V_y^T$ iff the columns of $\widehat{U}_y$ are in the span of $\widehat{U}_g$ as in*

$$\min_{W \in \mathbb{R}^{K \times C}} \|Y - Z^* W\|_F^2 = 0 \iff \widehat{U}_y \in \text{span}(\widehat{U}_g),$$

*with $Z^*$ the embeddings of the VICReg -or- SimCLR -or- BarlowTwins model after convergence. (Proof in Appendix F.16.)*

Although not explicitly stated in Theorem 9 the same applies e.g. to NNCLR and MeanShift as they employ the same SimCLR loss, only the design of $G$ is altered. The above is crucial in helping and guiding the design of SSL methods and theoretically confirm the empirical findings from Geirhos et al. [72] that observed in different scenarios that SSL and supervised models nearly fall back to the same thing.

## C.4   Non-Contrastive Methods Should be Preferred: Best and Worst Downstream Task Error Bounds

We demonstrated in Appendix C.3 that all SSL methods can be optimal to solve a task at ahdn as long as the spectral properties of $G$ and $Y$ are aligned. However, this is rarely the case in practical scenarios, and it thus becomes crucial

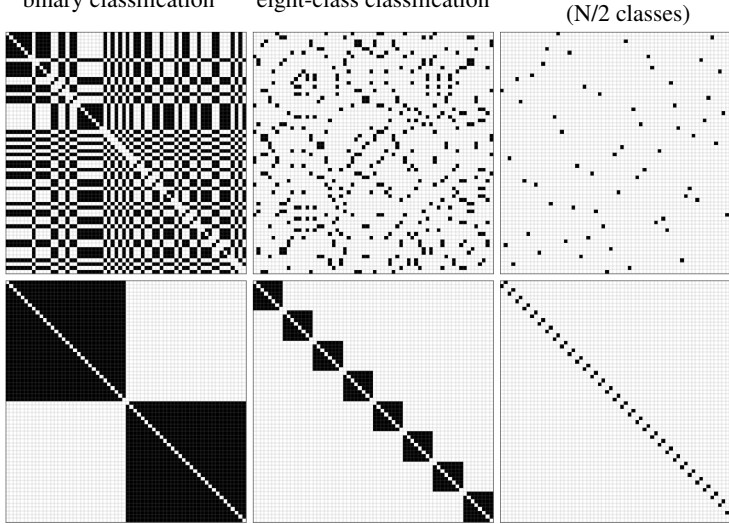

binary classification     eight-class classification     pairwise relation (N/2 classes)

Figure 6: Examples of the $N \times N$ symmetric adjacency matrices $\boldsymbol{G}$ encountered in supervised classification (**two left columns**) and in SSL (**right column**) with $N$ samples, in the latter case only pairwise relationships are known. We also present a random sample order (**top row**) and a per-class sample ordering (**bottom row**) to better visualize the structure of $\boldsymbol{G}$, such ordering permutation does not alter the results of this study. Each any case, entries $(\boldsymbol{G})_{i,j}$ represent the known positive relation (when nonzero) between samplee $i$ and $j$. The key insight that will play a key role in our analysis is that the eigenvectors of those matrices entirely encode the similarity information between samples and that SSL methods push $\boldsymbol{Z}$ left-singular vectors to match the ones of $\boldsymbol{G}$ (Theorems 1, 5 and 6). Although $\boldsymbol{G}$ is commonly symmetric, it is not always required (see e.g. Theorem 7).

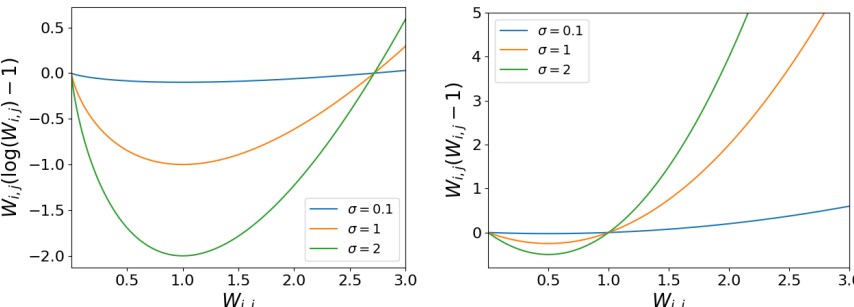

Figure 7: Depiction of the regularization term $\mathcal{R}_{\log}$ (**left**) and $\mathcal{R}_{\log}$ (**right**) from Eq. (11) for varying value of $(\boldsymbol{W})_{i,j}$ and three different temperatures $\tau \in \{0.1, 1, 2\}$ demonstrating how such regularization prevents $\widehat{\boldsymbol{G}}$ to collapse to the trivial 0 matrix in the optimization problem in Eq. (10).

to understand the behavior of the learned representation $\boldsymbol{Z}$ on downstream task and if it varies with different SSL methods. First, we propose the following bound which represent the best and worst case downstream performances as a function of the rank of $\boldsymbol{G}$ which is mostly a result of applying the Eckart-Young-Mirsky theorem. That is, we look at all the possible similarity matrices $\boldsymbol{G}$ of rank $R$ and see, given a task $\boldsymbol{Y}$ what is the best achievable performance if $\boldsymbol{G}$ correctly encoders the data geometry, and what is the worse possible performance if $\boldsymbol{G}$ is "orthogonal" to the correct data geometry. For clarity and without loss of generality we assume here that $\mathrm{rank}(\boldsymbol{Y}) \leq K$ as otherwise no method would produce an optimal representation in general and $K < N$ as otherwise we are in the kernel regime.

**Theorem 10.** *Given fixed inputs $\boldsymbol{X}$ the lower and upper-bound over all possible matrices $\boldsymbol{G}$ of rank $R$ of the downstream task performances (with fixed $\boldsymbol{Y}$) are given by*

$$\sum_{i=R}^{K} (\boldsymbol{\Sigma}_y^2)_{i,i} \leq \left( \min_{\boldsymbol{W} \in \mathbb{R}^{K \times C}} \mathcal{L}(\boldsymbol{W}, \boldsymbol{Z}_{\mathrm{BT/SimCLR}}^*(\boldsymbol{G})) - \min_{\boldsymbol{W} \in \mathbb{R}^{K \times C}} \mathcal{L}(\boldsymbol{W}, \boldsymbol{Z}_{\mathrm{VICReg}}^*(\boldsymbol{G})) \right) \leq \|\boldsymbol{Y}\|_F^2,$$

*and are tight. Hence one should prefer VICReg, then BarlowTwins and finally SimCLR to maximize the downstream task performances.*

The above result is a direct consequence of SimCLR forcing the representation to have the same rank as $\boldsymbol{G}$ while VICReg always enforce a full-rank representation. And although this difference becomes irrelevant with correct $\boldsymbol{G}$ (recall Appendix C.3) it becomes an important distinctive attribute between SSL methods when $\boldsymbol{G}$ is not optimal, which concerns most practical scenarios.

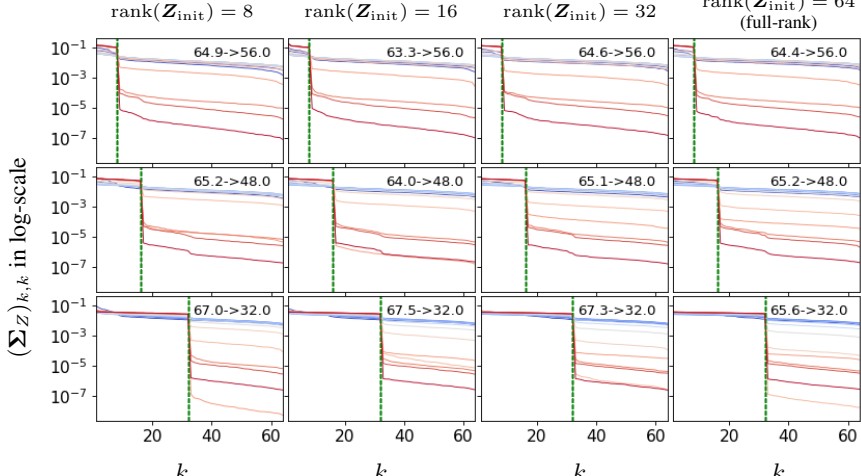

Figure 8: Depiction of the singular values $\boldsymbol{\Sigma}_z$ of the representation $\boldsymbol{Z}$ learned by the SimCLR/NNCLR loss (Eq. (2)) with varying similarity matrices $\boldsymbol{G}$ rank ($8, 16, 32$ **rows, green dotted lines**) during training (**from blue to red**) using different distances and similarity regularization (**columns**, recall Theorem 4) validating the result from Theorem 5 that the rank of the learned representation matches exactly the one of $\boldsymbol{G}$ which is a clear distinction between contrastive and non-constrastive losses.

## D  Additional Figures

## E  Exploration of SimCLR Variants

We now propose in this section to demonstrate how our findings i.e. obtaining the general optimization problem that SimCLR solves, can help us obtain novel and more principled SSL variants.

To do so, we propose to solve Eq. (10) but now with a different constraint than the one recovering SimCLR in Theorem 4. For example (derivations in Appendix F.9) the following variations can be obtained

$$\widehat{\boldsymbol{G}}_{d,\mathcal{R}_{\mathrm{F}}} = \mathrm{ReLU}\left(\boldsymbol{1}\boldsymbol{1}^T - \boldsymbol{I} - \boldsymbol{D}/\tau\right), \qquad (\text{with } \mathcal{G}, \tau > 0), \qquad (16)$$

$$\widehat{\boldsymbol{G}}_{d,\mathcal{R}_{\mathrm{F}}} = \mathrm{ReLU}\left(\frac{1}{N-1}(\boldsymbol{1}\boldsymbol{1}^T - \boldsymbol{I}) - \frac{1}{\tau}\boldsymbol{D}(\boldsymbol{I} - \frac{1}{N-1}\boldsymbol{1}\boldsymbol{1}^T)\right), \qquad (\text{with } \mathcal{G}_{\mathrm{rsto}}, \tau > 0), \qquad (17)$$

only by using a different regularizer. Depending on the application at hand, using the above formula as opposed to the usual SimCLR ones could prove beneficial. One simple scenario would be when the batch size is large i.e. the estimate of the $\boldsymbol{G}$ matrix has a quadratic memory complexity. Using the above we observed that thanks to the emergence of the ReLU activation (coming from the specific regularizer and its impact on the analytical graph estimate) the matrix estimate $\widehat{\boldsymbol{G}}_{d,\mathcal{R}_{\mathrm{F}}}$ is sparse. Hence it is now possible to tweak the temperature $\rho$ parameter to reach a desired level of sparsity while possibly preserving performances. Although this benefit might seem anecdotal, we believe that deriving novel estimates from a priori knowledge on the application at hand will produce more aligned SSL methods with further improving performances.

## F  Formal Statements Proofs

This section of the supplementary materials is providing the proofs of the main's paper formal results. We also provide as much background results and references as possible throughout to ensure that all the derivations are self-contained. Some of the below derivation do not belong to formal statements but are included to help the curious readers get additional insights into current SSL methods.

### F.1  VICReg Variance+Covariance Versus Representation's Singular Values

This simple derivation demonstrates how minimizing the VCReg can be done through an upper bound by constraining all the singular-values to be close to 1 although the general criterion only enforces for the variance term to be —at

least— 1 through the following derivations

$$\min_{\boldsymbol{Z}} \min_{\boldsymbol{u}\in[1,\infty)^K} \|\boldsymbol{Z}^T\boldsymbol{Z} - \mathrm{diag}(\boldsymbol{u})\|_F^2 = \min_{\boldsymbol{Z}} \min_{\boldsymbol{u}\in[1,\infty)^K} \|V_{\boldsymbol{Z}}\Sigma_{\boldsymbol{Z}}^2 V_{\boldsymbol{Z}}^T - \mathrm{diag}(\boldsymbol{u})\|_F^2$$
$$= \min_{\boldsymbol{Z}} \min_{\boldsymbol{u}\in[1,\infty)^K} \|\Sigma_{\boldsymbol{Z}}^2 - \mathrm{diag}(\boldsymbol{u})\|_F^2$$
$$= \min_{\boldsymbol{Z}} \min_{\boldsymbol{u}\in[1,\infty)^K} \|\boldsymbol{\sigma}_{\boldsymbol{Z}}^2 - \boldsymbol{u}\|_2^2$$
$$\leq \min_{\boldsymbol{Z}} \|\boldsymbol{\sigma}_{\boldsymbol{Z}}^2 - \boldsymbol{1}\|_2^2$$

where we denoted by $\boldsymbol{\sigma}_{\boldsymbol{Z}}$ the diagonal part of the diagonal $\Sigma_{\boldsymbol{Z}}$ matrix.

## F.2 Non-Unique Solution to Least-Square

The below derivation demonstrates that even in the representation $\boldsymbol{Z}$ (or any input matrix) is not full-rank, the least-square type of solution $\boldsymbol{W}$ to predict $\boldsymbol{Y} = \boldsymbol{Z}\boldsymbol{W}$ can be found, it is just not unique. In fact, it is possible to find an entire space of matrices that will minimize the loss i.e. respect the below equality that makes the loss have a gradient of $0$ and be at its minimum value, simply by moving within the kernel space of the representation (input) matrix as in

$$\boldsymbol{Z}^T\boldsymbol{Y} = \boldsymbol{Z}^T\boldsymbol{Z}\boldsymbol{W} \iff (U_{\boldsymbol{Z}}\Sigma_{\boldsymbol{Z}}V_{\boldsymbol{Z}}^\top)^T\boldsymbol{Y} = (U_{\boldsymbol{Z}}\Sigma_{\boldsymbol{Z}}V_{\boldsymbol{Z}}^\top)^\top(U_{\boldsymbol{Z}}\Sigma_{\boldsymbol{Z}}V_{\boldsymbol{Z}}^\top)\boldsymbol{W}$$
$$\iff V_{\boldsymbol{Z}}\Sigma_{\boldsymbol{Z}}^T U_{\boldsymbol{Z}}^T\boldsymbol{Y} = V_{\boldsymbol{Z}}\Sigma_{\boldsymbol{Z}}^T\Sigma_{\boldsymbol{Z}}V_{\boldsymbol{Z}}^\top\boldsymbol{W}$$
$$\iff \boldsymbol{W}^* \in \{(V_{\boldsymbol{Z}}\Sigma_{\boldsymbol{Z}}^{-1}U_{\boldsymbol{Z}}^T + \boldsymbol{M})\boldsymbol{Y} : \boldsymbol{M} \in \ker(\boldsymbol{Z})\},$$

where we recall that $\ker(\boldsymbol{Z})$ is the kernel space of $\boldsymbol{Z}$, and where we slightly abused notations by employing $\Sigma_{\boldsymbol{Z}}^{-1}$ to represent the inverse only for the non-zero element of the diagonal matrix $\Sigma_{\boldsymbol{Z}}$ (there are as many zeros in the diagonal as the dimension of $\ker(\boldsymbol{Z})$).

## F.3 Any Linear Weight from Appendix F.2 Has Zero Least-Square Gradient

This section continues the previous derivations but now demonstrating that using this optimal value for $\boldsymbol{W}$ denoted as $\boldsymbol{W}^*$ does fullfil the equality i.e. we are at a global optimum for any matrix within the defined subspace as in

$$V_{\boldsymbol{Z}}\Sigma_{\boldsymbol{Z}}^T U_{\boldsymbol{Z}}^T\boldsymbol{Y} = V_{\boldsymbol{Z}}\Sigma_{\boldsymbol{Z}}^T\Sigma_{\boldsymbol{Z}}V_{\boldsymbol{Z}}^\top\boldsymbol{W}^*$$
$$\implies V_{\boldsymbol{Z}}\Sigma_{\boldsymbol{Z}}^T U_{\boldsymbol{Z}}^T\boldsymbol{Y} = V_{\boldsymbol{Z}}\Sigma_{\boldsymbol{Z}}^T\Sigma_{\boldsymbol{Z}}V_{\boldsymbol{Z}}^\top(V_{\boldsymbol{Z}}\Sigma_{\boldsymbol{Z}}^{-1}U_{\boldsymbol{Z}}^T + \boldsymbol{M})\boldsymbol{Y}$$
$$\implies V_{\boldsymbol{Z}}\Sigma_{\boldsymbol{Z}}^T U_{\boldsymbol{Z}}^T\boldsymbol{Y} = V_{\boldsymbol{Z}}\Sigma_{\boldsymbol{Z}}^T\Sigma_{\boldsymbol{Z}}\Sigma_{\boldsymbol{Z}}^{-1}U_{\boldsymbol{Z}}^T\boldsymbol{Y}$$
$$\implies V_{\boldsymbol{Z}}\Sigma_{\boldsymbol{Z}}^T U_{\boldsymbol{Z}}^T\boldsymbol{Y} = V_{\boldsymbol{Z}}\,\mathrm{diag}(\sigma_{\boldsymbol{Z}})^2\Sigma_{\boldsymbol{Z}}^{-1}U_{\boldsymbol{Z}}^T\boldsymbol{Y}$$
$$\implies V_{\boldsymbol{Z}}\Sigma_{\boldsymbol{Z}}^T U_{\boldsymbol{Z}}^T\boldsymbol{Y} = V_{\boldsymbol{Z}}\Sigma_{\boldsymbol{Z}}^T U_{\boldsymbol{Z}}^T\boldsymbol{Y}$$

where the last equality follows since $\mathrm{diag}(\sigma_{\boldsymbol{Z}})^2\Sigma_{\boldsymbol{Z}}^{-1}$ will either multiply $(\sigma_{\boldsymbol{Z}})_i^2$ with $(\Sigma_{\boldsymbol{Z}}^{-1})_{i,i} = \frac{1}{(\sigma_{\boldsymbol{Z}})_i}$ if $(\sigma_{\boldsymbol{Z}})_i$ is nonzero, and otherwise if $(\sigma_{\boldsymbol{Z}})_i$ is 0, then it will be the product between 0 for $(\sigma_{\boldsymbol{Z}})_i^2$ and 0 for $(\Sigma_{\boldsymbol{Z}}^{-1})_{i,i}$ giving back the original value for $(\sigma_{\boldsymbol{Z}})_i$.

## F.4 Achievable Loss with Low-Rank Representation

This section takes a last detour towards understanding the least-square loss with a low-rank input/representation matrix. In this case we derive a various set of quantities that quantify the minimum loss that is achieved by any of the optimal matrix $\boldsymbol{W}^*$ found in Appendix F.2 as follows again slightly abusing notations for $\Sigma_{\boldsymbol{Z}}^{-1}$ to only invert the non-zero

singular values of $\boldsymbol{Z}$ as

$$
\begin{aligned}
\frac{1}{2}\|\boldsymbol{Y} - \boldsymbol{Z}\boldsymbol{W}^*\|_F^2 &= \frac{1}{2}\|\boldsymbol{Y} - \boldsymbol{Z}(V_{\boldsymbol{Z}}\Sigma_{\boldsymbol{Z}}^{-1}U_{\boldsymbol{Z}}^T + \tilde{V}_{\boldsymbol{Z}}\boldsymbol{M})\boldsymbol{Y}\|_F^2 \\
&= \frac{1}{2}\|\boldsymbol{Y} - U_{\boldsymbol{Z}}\Sigma_{\boldsymbol{Z}}V_{\boldsymbol{Z}}^\top V_{\boldsymbol{Z}}\Sigma_{\boldsymbol{Z}}^{-1}U_{\boldsymbol{Z}}^T\boldsymbol{Y}\|_F^2 \\
&= \frac{1}{2}\|\boldsymbol{Y} - U_{\boldsymbol{Z}}\Sigma_{\boldsymbol{Z}}\Sigma_{\boldsymbol{Z}}^{-1}U_{\boldsymbol{Z}}^T\boldsymbol{Y}\|_F^2 \\
&= \frac{1}{2}\|(\boldsymbol{I} - U_{\boldsymbol{Z}}\Sigma_{\boldsymbol{Z}}\Sigma_{\boldsymbol{Z}}^{-1}U_{\boldsymbol{Z}}^T)\boldsymbol{Y}\|_F^2 \\
&= \frac{1}{2}\|(U_{\boldsymbol{Z}}U_{\boldsymbol{Z}}^T - U_{\boldsymbol{Z}}\Sigma_{\boldsymbol{Z}}\Sigma_{\boldsymbol{Z}}^{-1}U_{\boldsymbol{Z}}^T)\boldsymbol{Y}\|_F^2 \\
&= \frac{1}{2}\|(\boldsymbol{I} - \Sigma_{\boldsymbol{Z}}\Sigma_{\boldsymbol{Z}}^{-1})U_{\boldsymbol{Z}}^T\boldsymbol{Y}\|_F^2 \\
&= \frac{1}{2}\|\boldsymbol{Y}\|_F^2 - \frac{1}{2}\|\Sigma_{\boldsymbol{Z}}\Sigma_{\boldsymbol{Z}}^{-1}U_{\boldsymbol{Z}}^T\boldsymbol{Y}\|_F^2 \\
&= \frac{1}{2}\|\boldsymbol{Y}\|_F^2 - \frac{1}{2}\|\Sigma_{\boldsymbol{Z}}\Sigma_{\boldsymbol{Z}}^{-1}U_{\boldsymbol{Z}}^T U_y\Sigma_y\|_F^2 \\
&= \frac{1}{2}\|\boldsymbol{Y}\|_F^2 - \frac{1}{2}\sum_{i,j}\mathbb{1}_{\{(\sigma_{\boldsymbol{Z}})_i>0\}}\langle(U_{\boldsymbol{Z}})_i, (U_y)_j\rangle^2(\sigma_y)_j^2,
\end{aligned}
$$

where it is clear that the minimum loss will in general not be 0 unless $\Sigma_{\boldsymbol{Z}}\Sigma_{\boldsymbol{Z}}^{-1} = \boldsymbol{I}$ which is not guaranteed (recall that we abuse notation for the inverse and that $\Sigma_{\boldsymbol{Z}}\Sigma_{\boldsymbol{Z}}^{-1}$ is only 1 in its diagonal for the nonzero singular values of $\boldsymbol{Z}$.

### F.5 Equivalence Between VICReg Invariance Term and Trace with Graph Laplacian

The goal of this section is to derive the first crucial result of our study that ties VICReg to spectral embedding methods by doing a first connection between VICReg invariance loss and the Dirichlet energy of a graph. The equality follows from the Laplacian of the graph definition $\boldsymbol{L} = \boldsymbol{D} - \boldsymbol{G}$ where $\boldsymbol{D}$ is the degree matrix of the graph i.e. a diagonal matrix with entries corresponding to the sum of each row of $\boldsymbol{G}$, and using the following algebraic manipulations which are common in the spectral graph analysis community, see e.g. [34]

$$
\begin{aligned}
\mathrm{Tr}\left(\boldsymbol{Z}^T\boldsymbol{L}\boldsymbol{Z}\right) &= \sum_d(\boldsymbol{Z}^T\boldsymbol{L}\boldsymbol{Z})_{d,d} = \sum_{i,j}\boldsymbol{L}_{i,j}\langle(\boldsymbol{Z})_{j,.}.(\boldsymbol{Z})_{i,.}\rangle \\
&= \sum_i\|(\boldsymbol{Z})_{i,.}\|_2^2(\boldsymbol{D})_{i,i} - \sum_{i,j}(\boldsymbol{G})_{i,j}\langle(\boldsymbol{Z})_{j,.}.(\boldsymbol{Z})_{i,.}\rangle \\
&= \frac{\sum_i\|(\boldsymbol{Z})_{i,.}\|_2^2(\boldsymbol{D})_{i,i} + \sum_i\|(\boldsymbol{Z})_{i,.}\|_2^2(\boldsymbol{D})_{i,i}}{2} - \sum_{i,j}(\boldsymbol{G})_{i,j}\langle(\boldsymbol{Z})_{j,.}.(\boldsymbol{Z})_{i,.}\rangle \\
&= \frac{\sum_i\sum_j\|(\boldsymbol{Z})_{i,.}\|_2^2(\boldsymbol{G})_{i,j} + \sum_i\sum_j\|(\boldsymbol{Z})_{i,.}\|_2^2(\boldsymbol{G})_{i,j}}{2} - \sum_{i,j}(\boldsymbol{G})_{i,j}\langle(\boldsymbol{Z})_{j,.}, (\boldsymbol{Z})_{i,.}\rangle \\
&= \frac{1}{2}\sum_i\sum_j(\boldsymbol{G})_{i,j}\|(\boldsymbol{Z})_{i,.} - (\boldsymbol{Z})_{j,.}\|_2^2,
\end{aligned}
$$

which is a famous derivation in graph signal processing relating pairwise distances to Dirichlet energy of the underlying graph with Laplacian $\boldsymbol{L}$.

### F.6 Non-Uniqueness of a Representation to a Given VICReg Loss Value

In this section we provide a simple argument to demonstrate that the VICReg representation that obtains a loss value of $c > 0$ is not unique for any positive value $c$.

The simplest solution to show this is to consider the operators involves in the forming of the VICReg loss. Looking at the variance and covariance terms $\mathcal{L}_{\mathrm{var}}, \mathcal{L}_{\mathrm{cov}}$ for example, it is direct to observe that $\mathcal{L}_{\mathrm{var}}(\boldsymbol{Z}) = \mathcal{L}_{\mathrm{var}}(\boldsymbol{Z} + \boldsymbol{C})$ and $\mathcal{L}_{\mathrm{cov}}(\boldsymbol{Z}) = \mathcal{L}_{\mathrm{cov}}(\boldsymbol{Z} + \boldsymbol{C})$ where $\boldsymbol{C}$ is a constant matrix in all its entries. Now, considering the invariance term $\mathcal{L}_{\mathrm{inv}}$,

one also obtains quite easily that

$$\mathcal{L}_{\text{inv}}(\boldsymbol{Z} + \boldsymbol{C}) = \sum_{i,j} \|(\boldsymbol{Z} + \boldsymbol{C})_{i,.} - (\boldsymbol{Z} + \boldsymbol{C})_{j,.} - \|$$

$$= \sum_{i,j} \|(\boldsymbol{Z})_{i,.} - (\boldsymbol{Z})_{j,.}\|$$

$$= \mathcal{L}_{\text{inv}}(\boldsymbol{Z}),$$

and thus, when looking at the entire VICReg loss, we see that it has the same value for $\boldsymbol{Z}$ and $\boldsymbol{Z} + \boldsymbol{C}$.

We thus obtain that for any reachable loss value $c$, the representation that achieves such loss is not unique —up to scalar shifting, concluding the proof.

### F.7 Optimal Representation and Loss for VICReg (Theorem 1)

The goal of this section is to obtain the closed-form optimal representation of VICReg using the least-square variance loss, instead of the hinge-loss, and to find the minimum loss associated to that (non-unique) optimum. Using the Trace term derivations given in Appendix F.5 we obtain

$$\mathcal{L}_{\text{VIC}} = \alpha \|\text{Cov}(\boldsymbol{Z}^*_{\alpha,\gamma}) - \boldsymbol{I}\|_F^2 + 2\frac{\gamma}{N} \text{Tr}\left( (\boldsymbol{P}'_{\alpha,\gamma}(\boldsymbol{\Lambda}'_{\alpha,\gamma}N)^{1/2})^T \boldsymbol{L} \boldsymbol{P}'_{\alpha,\gamma}(\boldsymbol{\Lambda}'_{\alpha,\gamma}N)^{1/2} \right),$$

where we use $\boldsymbol{\lambda}'_{\alpha,\beta}$, $\boldsymbol{P}'_{\alpha,\gamma}$ and $\boldsymbol{\Lambda}'_{\alpha,\gamma}$ to denote the first $K$ indices/columns of Eq. (7). Simplifying the trace term leads to

$$\frac{\gamma}{N} \text{Tr}\left( (\boldsymbol{P}'_{\alpha,\gamma}(\boldsymbol{\Lambda}'_{\alpha,\gamma}N)^{1/2})^T \boldsymbol{L} \boldsymbol{P}'_{\alpha,\gamma}(\boldsymbol{\Lambda}'_{\alpha,\gamma}N)^{1/2} \right) = \gamma \text{Tr}\left( \boldsymbol{\Lambda}'_{\alpha,\gamma} \boldsymbol{P}'^T_{\alpha,\gamma} \boldsymbol{L} \boldsymbol{P}'_{\alpha,\gamma} \right)$$

$$= \alpha \text{Tr}\left( \boldsymbol{\Lambda}'_{\alpha,\gamma} \boldsymbol{P}'^T_{\alpha,\gamma} \boldsymbol{M} \boldsymbol{P}'_{\alpha,\gamma} \right)$$

$$- \alpha \text{Tr}\left( \boldsymbol{\Lambda}'_{\alpha,\gamma} \boldsymbol{P}'^T_{\alpha,\gamma} (\boldsymbol{M} - \frac{\gamma}{\alpha}\boldsymbol{L}) \boldsymbol{P}'_{\alpha,\gamma} \right)$$

$$= \alpha \text{Tr}\left( \boldsymbol{\Lambda}'_{\alpha,\gamma} \boldsymbol{P}'^T_{\alpha,\gamma} (\boldsymbol{I} - \frac{1}{N}\boldsymbol{1}\boldsymbol{1}^T) \boldsymbol{P}'_{\alpha,\gamma} \right) - \alpha \|\boldsymbol{\lambda}'_{\alpha,\gamma}\|_2^2,$$

plugging this value into the loss we obtain

$$\mathcal{L}_{\text{VIC}} = \alpha \left( \|\text{Cov}(\boldsymbol{Z}^*_{\alpha,\gamma}) - \boldsymbol{I}\|_F^2 + 2\text{Tr}\left( \boldsymbol{\Lambda}'_{\alpha,\gamma} \boldsymbol{P}'^T_{\alpha,\gamma} (\boldsymbol{I} - \frac{1}{N}\boldsymbol{1}\boldsymbol{1}^T) \boldsymbol{P}'_{\alpha,\gamma} \right) - 2\|\boldsymbol{\lambda}'_{\alpha,\gamma}\|_2^2 \right)$$

$$= \alpha \left( \|\text{Cov}(\boldsymbol{Z}^*_{\alpha,\gamma})\|_F^2 + K - 2\text{Tr}(\text{Cov}(\boldsymbol{Z}^*_{\alpha,\gamma})) + 2\text{Tr}\left( \boldsymbol{\Lambda}'_{\alpha,\gamma} \boldsymbol{P}'^T_{\alpha,\gamma} (\boldsymbol{I} - \frac{1}{N}\boldsymbol{1}\boldsymbol{1}^T) \boldsymbol{P}'_{\alpha,\gamma} \right) - 2\|\boldsymbol{\lambda}'_{\alpha,\gamma}\|_2^2 \right)$$

$$= \alpha \left( \|\text{Cov}(\boldsymbol{Z}^*_{\alpha,\gamma})\|_F^2 + K - 2\|\boldsymbol{\lambda}'_{\alpha,\gamma}\|_2^2 \right)$$

$$= \alpha \left( \left\| \frac{1}{N}(\boldsymbol{Z}^*_{\alpha,\gamma})^T \boldsymbol{M} \boldsymbol{Z}^*_{\alpha,\gamma} \right\|_F^2 + K - 2\|\boldsymbol{\lambda}'_{\alpha,\gamma}\|_2^2 \right),$$

now one should recall that $\boldsymbol{Z}^*_{\alpha,\gamma}$ contains the eigenvectors of $\boldsymbol{M} - \frac{\gamma}{\alpha}\boldsymbol{L}$ and that each of those eigenvector that has nonzero singular value has 0 mean since

$$(\boldsymbol{M} - \frac{\gamma}{\alpha}\boldsymbol{L})\boldsymbol{v} = \lambda\boldsymbol{v}$$

$$\iff \boldsymbol{M}(\boldsymbol{I} - \frac{\gamma}{\alpha}\boldsymbol{L})\boldsymbol{M}\boldsymbol{v} = \lambda\boldsymbol{v} \qquad \text{(Laplacian rows/cols sum to 0)}$$

$$\implies \boldsymbol{1}^T\boldsymbol{M}(\boldsymbol{I} - \frac{\gamma}{\alpha}\boldsymbol{L})\boldsymbol{M}\boldsymbol{v} = \lambda\boldsymbol{1}^T\boldsymbol{v}$$

$$\implies 0 = \lambda\boldsymbol{1}^T\boldsymbol{v} \implies \boldsymbol{1}^T\boldsymbol{v} = 0 \qquad \text{(for any eigenvector } \boldsymbol{v} \text{ with } \lambda > 0)$$

hence we obtain the following simplifications

$$
\begin{aligned}
\mathcal{L}_{\mathrm{VIC}} =& \alpha \left( \left\| \frac{1}{N}(\boldsymbol{Z}_{\alpha,\gamma}^*)^T \boldsymbol{M} \boldsymbol{Z}_{\alpha,\gamma}^* \right\|_F^2 + K - 2\|\boldsymbol{\lambda}_{\alpha,\gamma}'\|_2^2 \right) \\
=& \alpha \left( \left\| \frac{1}{N}(\boldsymbol{Z}_{\alpha,\gamma}^*)^T \boldsymbol{Z}_{\alpha,\gamma}^* \right\|_F^2 + K - 2\|\boldsymbol{\lambda}_{\alpha,\gamma}'\|_2^2 \right) \\
=& \alpha \left( \|\boldsymbol{\lambda}_{\alpha,\gamma}'\|_2^2 + K - 2\|\boldsymbol{\lambda}_{\alpha,\gamma}'\|_2^2 \right) \\
=& \alpha(K - \|\boldsymbol{\lambda}_{\alpha,\gamma}'\|_2^2)
\end{aligned}
$$

which conludes the proof.

### F.8 Proof of Constrained VICReg Recovering Laplacian Eigenmaps (Theorem 2)

This section takes on proving the first key result of our study that thoroughly tie (constrained) VICReg to the known local spectral embedding method Laplacian Eigenmap. The only difference would be that in our case the graph $\boldsymbol{G}$ is given from an SSL viewpoint and not constructed from a k-NN graph and geodesic distance estimate as in LE.

There are two differences between the constrained VICReg problem

$$
\min_{\boldsymbol{Z} \in \mathbb{R}^{N \times K}} \mathcal{L}_{\mathrm{inv}} \text{ s.t. } \mathcal{L}_{\mathrm{var}} = 0, \mathcal{L}_{\mathrm{cov}} = 0
$$

and the LE problem which is

$$
\min_{\boldsymbol{Z} \in \mathbb{R}^{N \times K}} \mathrm{Tr}(\boldsymbol{Z}^T(\boldsymbol{D} - \boldsymbol{G})\boldsymbol{Z}) \text{ s.t. } \boldsymbol{Z}^T \boldsymbol{D} \boldsymbol{Z} = \boldsymbol{I}.
$$

The first is the difference in the term being minimized, and the second the constraints imposed on $\boldsymbol{Z}$.

**The first difference** can be handled quite easily following the derivations from Appendix F.5 that demonstrated that the invariance term of VICReg inv corresponds to the Trace term $\mathrm{Tr}(\boldsymbol{Z}^T(\boldsymbol{D} - \boldsymbol{G})\boldsymbol{Z})$ (times 2) that LE tries to minimize. Hence, we already obtain the following reformulation of our constrained VICReg problem

$$
\min_{\boldsymbol{Z} \in \mathbb{R}^{N \times K}} \mathrm{Tr}(\boldsymbol{Z}^T(\boldsymbol{D} - \boldsymbol{G})\boldsymbol{Z}) \text{ s.t. } \mathcal{L}_{\mathrm{var}} = 0, \mathcal{L}_{\mathrm{cov}} = 0
$$

**The second difference** is also quite direct to deal with. In fact, setting the constraint $\mathcal{L}_{\mathrm{var}} = 0, \mathcal{L}_{\mathrm{cov}} = 0$ is equivalent to the LE constraint $\boldsymbol{Z}^T \boldsymbol{D} \boldsymbol{Z} = \boldsymbol{I}$ as we now show. To see that, first notice (or recall from Appendix F.7) that the variance+covariance term can be expressed as $\mathcal{L}_{\mathrm{var}} + \mathcal{L}_{\mathrm{cov}} = \|\frac{1}{N}\boldsymbol{Z}^T \boldsymbol{M} \boldsymbol{Z} - \boldsymbol{I}\|_F^2$, with $\boldsymbol{M}$ the centering matrix. Also, setting $\mathcal{L}_{\mathrm{var}} = 0, \mathcal{L}_{\mathrm{cov}} = 0$ is equivalent to setting $\mathcal{L}_{\mathrm{var}} + \mathcal{L}_{\mathrm{cov}} = 0$ as each term is non-negative. Hence we obtain a further simplification of the constrained VICReg into

$$
\min_{\boldsymbol{Z} \in \mathbb{R}^{N \times K}} \mathrm{Tr}(\boldsymbol{Z}^T(\boldsymbol{D} - \boldsymbol{G})\boldsymbol{Z}) \text{ s.t. } \left\| \frac{1}{N}\boldsymbol{Z}^T \boldsymbol{M} \boldsymbol{Z} - \boldsymbol{I} \right\|_F^2 = 0.
$$

The only step remaining is to show the equivalence between imposing $\left\|\frac{1}{N}\boldsymbol{Z}^T \boldsymbol{M} \boldsymbol{Z} - \boldsymbol{I}\right\|_F^2 = 0$ and $\boldsymbol{Z}^T \boldsymbol{D} \boldsymbol{Z} = \boldsymbol{I}$. For this step, recall (i) that in all our SSL settings, $\boldsymbol{D} = \boldsymbol{I}d$ is isotropic and thus $\boldsymbol{Z}^T \boldsymbol{D} \boldsymbol{Z} = \boldsymbol{I} \iff \boldsymbol{Z}^T \boldsymbol{Z} = \frac{1}{d}\boldsymbol{I}$ and (ii) that the minimizer of LE consists in taking the $[2 : K + 1]$ eigenvectors of the Laplacian matrix $\boldsymbol{D} - \boldsymbol{G}$ which is equivalent to only considering the eigenvectors with zero-mean and is in fact also equivalent to adding the constraint $\boldsymbol{Z}\boldsymbol{1} = \boldsymbol{0}$ to the original LE problem. Because of that, we directly see that the search space for $\boldsymbol{Z}$ is so that $\boldsymbol{Z}^T \boldsymbol{M} \boldsymbol{Z} = \boldsymbol{Z}^T \boldsymbol{Z}$.

Hence, the above demonstrates how the constrained VICReg and LE solution coincide up to a global rescaling of all the entries of the solution $\boldsymbol{Z}$, concluding the proof.

### F.9 Proof of Laplacian estimation with contrastive learning Theorem 4

The first step of the proof consists in recovering the softmax with any metric $d$ that computes the distance (whatever distance desired) between pairs of inputs.

**Graph Laplacian Estimation recovers the step 1 of contrastive methods using $\mathcal{R}_{\log}$.** To prove Eq. (12) (the case with $\mathcal{G}$ and not $\mathcal{G}_{\text{rsto}}$ can be done similarly be removing the row-sum-to-one constraint), we need to solve the following optimization problem

$$\arg\min_{\boldsymbol{W}\in\mathcal{G}_{\text{rsto}}} \sum_{i\neq j} d(f_\theta(\boldsymbol{x}_i), f_\theta(\boldsymbol{x}_j))\boldsymbol{W}_{i,j} + \tau \sum_{i\neq j}\boldsymbol{W}_{i,j}(\log(\boldsymbol{W}_{i,j})-1),$$

which we solve by introducing the constraint in the optimization problem with the Lagrangian to

$$\mathcal{L} = \sum_{i\neq j} d(f_\theta(\boldsymbol{x}_i), f_\theta(\boldsymbol{x}_j))\boldsymbol{W}_{i,j} + \tau \sum_{i\neq j}\boldsymbol{W}_{i,j}(\log(\boldsymbol{W}_{i,j})-1) + \sum_{i=1}^{N}\lambda_i(\sum_{j\neq i}\boldsymbol{W}_{i,j}-1) + \sum_i \beta_i \boldsymbol{W}_{i,i}$$

$$\implies \frac{\partial\mathcal{L}}{\partial\boldsymbol{W}_{i,j}} = \begin{cases} d(f_\theta(\boldsymbol{x}_i), f_\theta(\boldsymbol{x}_j)) + \tau\log(\boldsymbol{W}_{i,j}) + \lambda_i \iff i\neq j \\ \beta_i \iff i=j \end{cases}$$

$$\implies \frac{\partial\mathcal{L}}{\partial\lambda_i} = \sum_{j\neq i}\boldsymbol{W}_{i,j} - 1$$

$$\implies \frac{\partial\mathcal{L}}{\partial\beta i} = \boldsymbol{W}_{i,i},$$

we first solve for $\frac{\partial\mathcal{L}}{\partial\boldsymbol{W}_{i,j}} = 0$ for $i\neq j$ to obtain an expression of $\boldsymbol{W}_{i,j}$ as a function of $\lambda_i$ as

$$\frac{\partial\mathcal{L}}{\partial\boldsymbol{W}_{i,j}} = 0 \iff d(f_\theta(\boldsymbol{x}_i), f_\theta(\boldsymbol{x}_j)) + \tau\log(\boldsymbol{W}_{i,j}) + \lambda_i = 0$$

$$\iff \tau\log(\boldsymbol{W}_{i,j}) = -d(f_\theta(\boldsymbol{x}_i), f_\theta(\boldsymbol{x}_j)) - \lambda_i$$

$$\iff \boldsymbol{W}_{i,j} = e^{\frac{-1}{\tau}(d(f_\theta(\boldsymbol{x}_i), f_\theta(\boldsymbol{x}_j))+\lambda_i)}$$

$$\iff \boldsymbol{W}_{i,j} = e^{\frac{-1}{\tau}d(f_\theta(\boldsymbol{x}_i), f_\theta(\boldsymbol{x}_j))}e^{\frac{-1}{\tau}\lambda_i},$$

and now using $\frac{\partial\mathcal{L}}{\partial\lambda_i} = 0$ we will be able to solve for $\lambda_i$ as follows

$$\frac{\partial\mathcal{L}}{\partial\lambda_i} = 0 \iff \sum_{j\neq i}\boldsymbol{W}_{i,j} - 1 = 0$$

$$\iff e^{\frac{-\lambda_i}{\tau}}\sum_{j\neq i}e^{\frac{-1}{\tau}d(f_\theta(\boldsymbol{x}_i), f_\theta(\boldsymbol{x}_j))} = 1$$

$$\iff e^{\frac{-\lambda_i}{\tau}} = \frac{1}{\sum_{j\neq i}e^{\frac{-1}{\tau}d(f_\theta(\boldsymbol{x}_i), f_\theta(\boldsymbol{x}_j))}},$$

which allows us to finally obtain an explicit solution for $\frac{\partial\mathcal{L}}{\partial\boldsymbol{W}_{i,j}} = 0$ that does not depend on $\lambda_i$ as follows

$$\frac{\partial\mathcal{L}}{\partial\boldsymbol{W}_{i,j}} = 0 \iff \boldsymbol{W}_{i,j} = e^{\frac{-1}{\tau}d(f_\theta(\boldsymbol{x}_i), f_\theta(\boldsymbol{x}_j))}e^{\frac{-\lambda_i}{\tau}}$$

$$\iff \boldsymbol{W}_{i,j} = \frac{e^{\frac{-1}{\tau}d(f_\theta(\boldsymbol{x}_i), f_\theta(\boldsymbol{x}_j))}}{\sum_{j\neq i}e^{\frac{-1}{\tau}d(f_\theta(\boldsymbol{x}_i), f_\theta(\boldsymbol{x}_j))}}$$

which holds for $i\neq j$. Now for the case $i=j$ simply using the constraint and solving for $\beta_i$ directly gives $\boldsymbol{W}_{i,i} = 0$. Notice that we did not enforce the $\boldsymbol{W}_{i,j} = \boldsymbol{W}_{j,i}$ constraint, however, the optimum found fulfills it and thus we are in a scenario akin to undirected graph Laplacian estimation.

**Recovering SimCLR with Cosine Similarity.** Now, since we are using the cosine distance, defined as $f(\boldsymbol{x}, \boldsymbol{y}) = 1 - \frac{\langle\boldsymbol{x},\boldsymbol{y}\rangle}{\|\boldsymbol{x}\|\|\boldsymbol{y}\|}$, we can plug it in the above to finally obtain

$$\boldsymbol{W}_{i,j} = \frac{e^{\frac{-1}{\tau}d(f_\theta(\boldsymbol{x}_i), f_\theta(\boldsymbol{x}_j))}}{\sum_{j\neq i}e^{\frac{-1}{\tau}d(f_\theta(\boldsymbol{x}_i), f_\theta(\boldsymbol{x}_j))}}1_{\{i\neq j\}} = \frac{e^{\frac{1}{\tau}\frac{\langle f_\theta(\boldsymbol{x}_i), f_\theta(\boldsymbol{x}_j)\rangle}{\|f_\theta(\boldsymbol{x}_i)\|_2\|f_\theta(\boldsymbol{x}_j)\|_2}}}{\sum_{j\neq i}e^{\frac{1}{\tau}\frac{\langle f_\theta(\boldsymbol{x}_i), f_\theta(\boldsymbol{x}_j)\rangle}{\|f_\theta(\boldsymbol{x}_i)\|_2\|f_\theta(\boldsymbol{x}_j)\|_2}}}1_{\{i\neq j\}},$$

which is exactly the features used by SimCLR or NNCLR. The last step is direct, take the graph of known positive pairs of nearest neighbor, apply the cross entropy between those and the above, and one obtains that Eq. (12) recovers exactly the loss of those models.

**Graph Laplacian Estimation recovers the step 1 of contrastive methods using $\mathcal{R}_F$.** We will be using the $\mathcal{G}_{\mathrm{rsto}}$ space again as the case for $\mathcal{G}$ can be obtained easily by removing the Lagrangian constraint. we need to solve the following optimization problem

$$\arg\min_{\boldsymbol{W}\in\mathcal{G}_{\mathrm{rsto}}} \sum_{i\neq j} d(f_\theta(\boldsymbol{x}_i), f_\theta(\boldsymbol{x}_j))\boldsymbol{W}_{i,j} + \tau \sum_{i\neq j} \boldsymbol{W}_{i,j}(\boldsymbol{W}_{i,j}/2 - 1),$$

which is augmented with the constraints to

$$\mathcal{L} = \sum_{i\neq j} d(f_\theta(\boldsymbol{x}_i), f_\theta(\boldsymbol{x}_j))\boldsymbol{W}_{i,j} + \tau \sum_{i\neq j} \boldsymbol{W}_{i,j}(\boldsymbol{W}_{i,j}/2 - 1) + \sum_{i=1}^{N} \lambda_i(\sum_{j\neq i}\boldsymbol{W}_{i,j} - 1) + \sum_i \beta_i \boldsymbol{W}_{i,i}$$

which we will differentiate with respect to $\boldsymbol{W}, \lambda, \beta$ to obtain given $\mathcal{L}$ above

$$\implies \frac{\partial\mathcal{L}}{\partial\boldsymbol{W}_{i,j}} = \begin{cases} d(f_\theta(\boldsymbol{x}_i), f_\theta(\boldsymbol{x}_j)) + \tau\boldsymbol{W}_{i,j} - \tau + \lambda_i \iff i\neq j \\ \beta_i \iff i = j \end{cases}$$

$$\implies \frac{\partial\mathcal{L}}{\partial\lambda_i} = \sum_{j\neq i}\boldsymbol{W}_{i,j} - 1$$

$$\implies \frac{\partial\mathcal{L}}{\partial\beta i} = \boldsymbol{W}_{i,i},$$

setting $\frac{\partial\mathcal{L}}{\partial\boldsymbol{W}_{i,j}}$ to 0 we will obtain the following simplification isolating $\lambda_i$ for $i\neq j$

$$\frac{\partial\mathcal{L}}{\partial\boldsymbol{W}_{i,j}} = 0 \iff d(f_\theta(\boldsymbol{x}_i), f_\theta(\boldsymbol{x}_j)) + \tau\boldsymbol{W}_{i,j} - \tau + \lambda_i = 0$$

$$\iff \boldsymbol{W}_{i,j} = 1 - \frac{1}{\tau}\Big(d(f_\theta(\boldsymbol{x}_i), f_\theta(\boldsymbol{x}_j)) + \lambda_i\Big)$$

and now using $\frac{\partial\mathcal{L}}{\partial\lambda_i} = 0$ we will obtain

$$\frac{\partial\mathcal{L}}{\partial\lambda_i} = 0 \iff \sum_{j\neq i}\boldsymbol{W}_{i,j} - 1 = 0$$

$$\iff \sum_{j\neq i}\left(1 - \frac{1}{\tau}\Big(d(f_\theta(\boldsymbol{x}_i), f_\theta(\boldsymbol{x}_j)) + \lambda_i\Big)\right) - 1 = 0$$

$$\iff N - 1 + \frac{-\sum_{j\neq i} d(f_\theta(\boldsymbol{x}_i), f_\theta(\boldsymbol{x}_j))}{\tau} - \frac{(N-1)}{\tau}\lambda_i - 1 = 0$$

$$\iff \lambda_i = \frac{-\sum_{j\neq i} d(f_\theta(\boldsymbol{x}_i), f_\theta(\boldsymbol{x}_j))}{(N-1)} + \tau(1 - \frac{1}{N-1})$$

and then plugging that into the original system of equation leads to

$$\frac{\partial\mathcal{L}}{\partial\boldsymbol{W}_{i,j}} = 0 \iff \boldsymbol{W}_{i,j} = 1 - \frac{1}{\tau}\Big(d(f_\theta(\boldsymbol{x}_i), f_\theta(\boldsymbol{x}_j)) + \lambda_i\Big)$$

$$\iff \boldsymbol{W}_{i,j} = 1 - \frac{1}{\tau}\left(d(f_\theta(\boldsymbol{x}_i), f_\theta(\boldsymbol{x}_j)) + \Big(\frac{-\sum_{j\neq i} d(f_\theta(\boldsymbol{x}_i), f_\theta(\boldsymbol{x}_j))}{(N-1)} + \tau\big(1 - \frac{1}{N-1}\big)\Big)\right)$$

$$\iff \boldsymbol{W}_{i,j} = \frac{1}{N-1} - \frac{1}{\tau}\left(d(f_\theta(\boldsymbol{x}_i), f_\theta(\boldsymbol{x}_j)) - \frac{\sum_{j\neq i} d(f_\theta(\boldsymbol{x}_i), f_\theta(\boldsymbol{x}_j))}{(N-1)}\right)$$

for any $i\neq j$. Again solving for $\beta_i$ will lead directly to $\boldsymbol{W}_{i,i} = 0$. In matrix form, we thus obtain $\boldsymbol{W} = \frac{1}{N-1}(\mathbf{1}\mathbf{1}^T - \boldsymbol{I}) - \frac{1}{\tau}\Big(\boldsymbol{D} - \frac{1}{N-1}\boldsymbol{D}\mathbf{1}\mathbf{1}^T\Big)$ since the diagonal elements of $\boldsymbol{D}$ are 0.

### F.10    Proof of SimCLR Theorem 5

The goal of this section is to demonstrate that SimCLR, although optimizing the graph estimate $\widehat{G}$ to match $G$ forces the representation $Z$ to match $G$ through its outer-product. This result closely follows a multidimensional scaling type of reasoning which should not be a surprise being a global spectral embedding method. Let's say that $\tau$ is for example $1$ and that all distances are between $0$ and $1$ e.g. using the cosine distance to streamline the derivations (if not, simply set $\tau$ to the maximum value present in the matrix $D$). We thus obtain

$$\|G - \widehat{G}\|_F^2 = \|(\mathbf{1}\mathbf{1}^T - I - G) - ((\mathbf{1}\mathbf{1}^T - I - \widehat{G}))\|_F^2 = \|\underbrace{(\mathbf{1}\mathbf{1}^T - I - G)}_{\triangleq D'} + D)\|_F^2$$

and now we will basically decompose the loss into two orthogonal terms as follows using the centering matrix i.e. Householder transformation $H = I - \frac{1}{N}\mathbf{1}\mathbf{1}^T$ and noting that $H\mathbf{1} = \mathbf{0}$. We simplify

$$\|D' - D\|_F^2 = \left\|(H + \frac{1}{N}\mathbf{1}\mathbf{1}^T)(D' - D)(H + \frac{1}{N}\mathbf{1}\mathbf{1}^T)\right\|_F^2 \quad \text{(since } H + \frac{1}{N}\mathbf{1}\mathbf{1}^T = I)$$

$$= \|H(D' - D)H\|_F^2 + 2\operatorname{Tr}\left(H(D' - D)\frac{1}{N}\mathbf{1}\mathbf{1}^T\right) + \left\|\frac{1}{N}\mathbf{1}\mathbf{1}^T(D' - D)\frac{1}{N}\mathbf{1}\mathbf{1}^T\right\|_F^2$$

$$= \|H(D' - D)H\|_F^2 + \left\|\frac{1}{N}\mathbf{1}\mathbf{1}^T(D' - D)\frac{1}{N}\mathbf{1}\mathbf{1}^T\right\|_F^2$$

$$= \|H(D' - D)H\|_F^2 + (\operatorname{mean}(D) - \operatorname{mean}(D'))^2$$

$$= 4\left\|-\frac{1}{2}HD'H - (-\frac{1}{2}HDH)\right\|_F^2 + (\operatorname{mean}(D) - \operatorname{mean}(D'))^2$$

$$= 4\left\|-\frac{1}{2}HD'H - HZZ^TH\right\|_F^2 + (\operatorname{mean}(D) - \operatorname{mean}(D'))^2$$

$$= 4\left\|H(G + I - 2ZZ^T)H\right\|_F^2 + (\operatorname{mean}(D) - \operatorname{mean}(D'))^2$$

which can then be minimized using [73] to obtain that $Z$ will be proportional to the first $K$ eigenvectors of $G + I$ rescaled by the square root of their eigenvalues divided by $2\tau$. For the case with softmax and cross-entropy, we entirely leverage on Corollary 2.2 of Ganea et al. [74] (a study done in the context of the softmax dimensional bottleneck [75]) where it was shown that the minimum of the cross entropy loss is obtained whenever the rank of the pre-activations (logits) of the true and predicted probabilities have same rank. As soon as one uses an $\epsilon$ label-smoothing, to ensure that the logits of the distribution (rows of $G$) do not go to infinity in the case where a row contains only a single $1$, we obtain the desired result. For the RBF case, it is proven to be always nonsingular, regardless of the value of the temperature parameter, as long as the samples are all distinct [76], hence in our case, the only solution to minimize the loss is for the samples (features maps in this case) to collapse i.e. for the rank of $Z$ to align with the one of $G$, see Wathen and Zhu [77] for other kernels.

### F.11    SimCLR/NNCLR recovers ISOMAP and MDS (Proposition 2)

This proof essentially follows the same derivations than Appendix F.10 up until the last equality. Then, one can see that the eigenvectors of $G$ and $G + I$ are identical (only the eigenvalues are shifted by 1). Hence, up to a rescaling, solving the ISOMAP optimization problem or the SimCLR one are equivalent, up to shift and rescaling of the representations.

### F.12    Barlow Twins optimal Representation (Theorem 6)

Canonical correlation analysis usually takes the form of an optimization problem searching over linear weights $W$ that maximally correlates the transformed inputs. In our case, we ought to work directly with the representation as it is the final input representation that is being fed into the BarlowTwins loss. To ease this proof, we will heavily rely on the derivations from Appendix F.15 since it offers the exact link between BarlowTwins and Canonical Correlation Analysis. The kernel version is nothing more that linear CCA but with input the $\phi(x_n)$ representations as opposed to the actual inputs $x_n$. Hence Appendix F.15 can be used in the same way to obtain that BarlowTwins with a nonlinear DN recovers Kernel CCA. Now, the key result concerns the rank of the representation. First, we ought to recall that we are working with the regularized version of CCA that adds a small constant to the denominator of the cosine similarity computation between the columns of $Z_a$ and $Z_b$.

### F.13 Linear VICReg is Locality Preserving Projection (Theorem 3)

Given the assumption that the within-cluster variance is positive for all the clusters, we have that $\boldsymbol{X}^T(\boldsymbol{D} - \boldsymbol{G})\boldsymbol{X}$ is invertible. Note that if this is not the case, one can use a result from [78] to obtain the same conclusion, although omitted in this study. Given that, it is will known that Laplacian Eigenmap with linear mapping of the input produces LPP, hence the proof essentially relies on Appendix F.8 that tied VICReg with Laplacian Eigenmap, and then e.g. on [79] for that known relationship between those two models.

### F.14 Linear VICReg with supervised relation matrix is LDA (Theorem 3)

We should highlight that the exact LDA employs an optimization problem that is typically nonconvex, and for which there does not exist a closed-form solution [80]. Hence it is common to look at a simpler optimization problem instead [81, 82] which is the one that VICReg closed-form solution recovers. For this section and without loss of generation we simply our notations by assuming that $\boldsymbol{G}$ as a degree matrix of $\boldsymbol{I}$ i.e. the sum of each row sum to one. The objective in 2-class LDA (see Li and Wang [83] for examples) is to minimize the following objective

$$\max \frac{\boldsymbol{w}^T \boldsymbol{\Sigma}_b \boldsymbol{w}}{\boldsymbol{w}^T \boldsymbol{\Sigma}_w \boldsymbol{w}},$$

where $\boldsymbol{\Sigma}_t = \boldsymbol{X}^T \boldsymbol{X}, \boldsymbol{\Sigma}_w = \boldsymbol{X}^T(\boldsymbol{I} - \boldsymbol{G})\boldsymbol{X}, \boldsymbol{\Sigma}_b = \boldsymbol{X}^T \boldsymbol{G} \boldsymbol{X}$ encode the total/within/between cluster variances respectively using the supervised relation matrix $\boldsymbol{G}$. In fact, (assuming for clarity that the inputs have $0$ mean) one has

$$
\begin{aligned}
\frac{1}{N}\boldsymbol{X}^T \boldsymbol{G} \boldsymbol{X} &= \frac{1}{N} \sum_{c=1}^{C} \sum_{i \in \mathcal{N}_c} \sum_{j \in \mathcal{N}_c} \frac{1}{N_c} \boldsymbol{x}_i \boldsymbol{x}_j^T \\
&= \frac{1}{N} \sum_{c=1}^{C} N_c \sum_{i \in \mathcal{N}_c} \sum_{j \in \mathcal{N}_c} \frac{1}{N_c^2} \boldsymbol{x}_i \boldsymbol{x}_j^T \\
&= \frac{1}{N} \sum_{c=1}^{C} N_c \frac{\sum_{i \in \mathcal{N}_c} \boldsymbol{x}_i}{N_c} \frac{\sum_{j \in \mathcal{N}_c} \boldsymbol{x}_j^T}{N_c} \\
&= \sum_{c=1}^{C} \frac{N_c}{N} \boldsymbol{\mu}_c \boldsymbol{\mu}_c^T,
\end{aligned}
$$

recovering the between (inter-cluster) variance $\boldsymbol{\Sigma}_b$ with $\boldsymbol{\mu}_c$ the center of class $c$. Now, since we have that $\boldsymbol{\Sigma}_t = \boldsymbol{\Sigma}_b + \boldsymbol{\Sigma}_w$ we directly have that $\boldsymbol{\Sigma}_w = \frac{1}{N}\left(\boldsymbol{X}^T \boldsymbol{X} - \boldsymbol{X}^T \boldsymbol{G} \boldsymbol{X}\right) = \frac{1}{N}\left(\boldsymbol{X}^T (\boldsymbol{I} - \boldsymbol{G}) \boldsymbol{X}\right)$. Given that, one can directly solve the LDA problem (in this case we provide directly the multivariate setting) via

$$\max_{\boldsymbol{W}} \frac{|\boldsymbol{W}^T \boldsymbol{\Sigma}_b \boldsymbol{W}|}{|\boldsymbol{W}^T \boldsymbol{\Sigma}_w \boldsymbol{W}|},$$

which is known as the Fisher criterion. Whenever $\boldsymbol{\Sigma}_w$ is invertible (if not, then the original data can be projected to a lower dimensional subspace without loss on information that would ensure that $\boldsymbol{\Sigma}_w$ is invertible) then Fisher's criterion is maximized leading to $\boldsymbol{W}$ being the solution to the generalized eigenvalue problem

$$\boldsymbol{W}^* = \arg\max \frac{|\boldsymbol{W}^T \boldsymbol{\Sigma}_b \boldsymbol{W}|}{|\boldsymbol{W}^T \boldsymbol{\Sigma}_w \boldsymbol{W}|} = \text{top eigenvectors of: } (\boldsymbol{\Sigma}_w)^{-1} \boldsymbol{\Sigma}_b.$$

Notice that the generalized eigenvalue problem is exactly

$$\boldsymbol{X}^T \boldsymbol{G} \boldsymbol{X} \boldsymbol{W} = \text{diag}(\lambda) \boldsymbol{X}^T (\boldsymbol{I} - \boldsymbol{G}) \boldsymbol{X} \boldsymbol{W},$$

and that $\boldsymbol{G}\boldsymbol{1} = \boldsymbol{1}$ we observe that this is exactly the same solution than the LPP problem with $\boldsymbol{G}$ being the relation matrix and $\boldsymbol{I}$ being the degree matrix of the graph. Hence we obtain that LPP is equivalent to LDA whenever one uses the supervised relation matrix $\boldsymbol{G}$ as $\boldsymbol{G}$ (which was first pointed out in [79]) and that it corresponds exactly to VICReg in the linear regime (as per Appendix F.13). We also note that the eigenvalue of the above matrix $(\boldsymbol{\Sigma}_w)^{-1}\boldsymbol{\Sigma}_b$ might seem to be arbitrary. However we have the following lemma that ensure that the eigenvalues of are real nonnegative.

**Lemma 1.** *The eigenvalues of $(\boldsymbol{\Sigma}_w)^{-1}\boldsymbol{\Sigma}_b$ are equal to the eigenvalues of $\boldsymbol{\Sigma}_w^{-1/2}\boldsymbol{\Sigma}_b\boldsymbol{\Sigma}_w^{-1/2}$ which is symmetric hence all are nonegative and real.*

*Proof.* The proof of the above statement comes from the fact that the eigenvalues of a product of matrices does not change under cyclic permutation of the matrix products. In fact, notice in general that

$$A_1 A_2 \ldots A_K v = \lambda v \iff A_K A_1 A_2 \ldots A_{K-1}(A_K v) = \lambda(A_K v), \tag{18}$$

where the $\iff$ holds as long as $A_K$ is invertible, and is otherwise $\implies$. The latter is sufficient since we only focus on the nonzero eigenvalues. Hence although the eigenvectors change when considering the eigendecomposition of $A_1 A_2 \ldots A_K$ as opposed to $A_K A_1 A_2 \ldots A_{K-1}$, the (nonzero) eigenvalues remain the same. Given this, it now becomes direct to apply this result onto $(\Sigma_w)^{-1}\Sigma_b = (\Sigma_w)^{-1/2}(\Sigma_w)^{-1/2}\Sigma_b$ to obtain the this matrix has the same eigenvalues than the matrix $(\Sigma_w)^{-1/2}\Sigma_b(\Sigma_w)^{-1/2}$. Now, because $\Sigma_b$ is positive semidefinite, we have that

$$v^T(\Sigma_w)^{-1/2}\Sigma_b(\Sigma_w)^{-1/2}v = ((\Sigma_w)^{-1/2}v)^T\Sigma_b((\Sigma_w)^{-1/2}v) \geq 0,$$

since $(\Sigma_w)^{-1/2}$ is symmetric. Hence, the eigenvalues of $(\Sigma_w)^{-1}\Sigma_b$ are nonnegative. $\square$

### F.15 Proof of BarlowTwins in the Linear Regime (Theorem 7)

The tie between CCA and LDA is not new. In fact, if one considers one dataset to be the samples and the other to be the class labels (binary variables for two-class problems or a variation of one-hot encoding for multi-class problems) then CCA and LDA are equivalent [84, 85]. But in our case, we do not explicitly use the labels but the views of different class. Hence, the goal of this section is to first recover the analytical parameters of BarlowTwins in the linear regime through CCA and then to demonstrate that the CCA-LDA connection also persists in the case when the views are both obtained from per-class samples i.e. exactly fitting the BarlowTwins scenario.

There exists many different ways to formulate the CCA problem. At the most simple level, one aims to sequentially learn pairs of filters that produce maximally correlated features as in

$$\max_{w_a, w_b} \frac{\langle X_a W_a, X_b W_b \rangle}{\|X_a W_a\|_2 \|X_b W_b\|_2},$$

which can easily be recognized to be the diagonal element of the BarlowTwins loss that we aim to maximize in the linear regime i.e. the cosine similarity between a column of $Z_a$ and the same column of $Z_b$. To find the corresponding filters, which are in general not assume to be identical, one transforms the above problem to a constrained optimization problem exactly as done with eigenvalue problems with the Rayleigh quotient. We thus obtain the following

$$\max_{w_a, w_b} \langle X_a W_a, X_b W_b \rangle \text{ s.t. } \|X_a W_a\|_2 = 1 \text{ and } \|X_b W_b\|_2 = 1,$$

since rescaling of the weights does not impact the learned filters, which can be transformed into a Lagrangian function as

$$\mathcal{L} = \langle X_a W_a, X_b W_b \rangle - \lambda_1(\|X_a W_a\|_2 - 1) - \lambda_2(\|X_b W_b\|_2 - 1),$$

which has the following partial derivatives using the $\Sigma_{ab}$ notations for the cross-products

$$\nabla_{w_a}\mathcal{L} = \Sigma_{ab} w_b - 2\lambda_1 \Sigma_{aa} w_a$$
$$\nabla_{w_b}\mathcal{L} = \Sigma_{ba} w_a - 2\lambda_2 \Sigma_{bb} w_b$$
$$\frac{\partial \mathcal{L}}{\partial \lambda_1} = w_a^T \Sigma_{aa} w_a$$
$$\frac{\partial \mathcal{L}}{\partial \lambda_2} = w_b^T \Sigma_{bb} w_b,$$

the first thing to notice is that setting those (first two) to zero will lead to $\lambda_1 = \lambda_2$ which we thus denote as a single $\lambda$ parameter since

$$\Sigma_{ab} w_b - 2\lambda_1 \Sigma_{aa} w_a = 0 \implies w_a^T \Sigma_{ab} w_b - 2\lambda_1 w_a^T \Sigma_{aa} w_a = 0 \implies \lambda_1 = \frac{1}{2} w_a^T \Sigma_{ab} w_b$$

$$\Sigma_{ba} w_a - 2\lambda_2 \Sigma_{bb} w_b = 0 \implies w_b^T \Sigma_{ba} w_a - 2\lambda_2 w_b^T \Sigma_{bb} w_b = 0 \implies \lambda_2 = \frac{1}{2} w_b^T \Sigma_{ba} w_a,$$

and thus we can simplify the original system using a single Lagrangian multiplier to have th following system

$$\left. \begin{array}{l} \Sigma_{ab} w_b - 2\lambda \Sigma_{aa} w_a = 0 \\ \Sigma_{ba} w_a - 2\lambda \Sigma_{bb} w_b = 0 \end{array} \right\} \qquad \left. \begin{array}{l} w_a = \frac{1}{2\lambda} \Sigma_{aa}^{-1} \Sigma_{ab} w_b \\ (\frac{1}{4}\Sigma_{bb}^{-1}\Sigma_{ba}\Sigma_{aa}^{-1}\Sigma_{ab} - \lambda^2 I) w_b = 0 \end{array} \right\},$$

where it is clear that one obtains $\boldsymbol{w}_{\mathrm{b}}$ by solving the generalized eigenvalue problem and then finds $\boldsymbol{w}_{\mathrm{a}}$ accordingly and this can be done by remove the $\frac{1}{4}$ factor and replacing $2\lambda$ by $\lambda$. For specific implementations of the above, we direct the reader to Healy [86], Ewerbring and Luk [87], Hardoon et al. [88].

**Recovering LDA from supervised CCA.** To recover this result, we closely follow the methodology from Kursun et al. [89]. Additionally, we obtain that in this case the two views are permutation of each other due to the symmetric structure of $\boldsymbol{G}$ that simply shuffles around the (repeated) samples to consider all the within-class pairs. Recall that even in t his case $\boldsymbol{G}$ as for degree matrix $\boldsymbol{I}$ since we only use positive pairs between the samples (the dataset has been augmented first duplicating each input in each class to match with all others). In that setting, we first have the following simplifications We obtain

$$\boldsymbol{\Sigma}_{aa} = \boldsymbol{Z}^T\boldsymbol{Z}, \;\; \boldsymbol{\Sigma}_{ab} = \boldsymbol{Z}^T\boldsymbol{G}\boldsymbol{Z}, \;\; \boldsymbol{\Sigma}_{ba} = \boldsymbol{Z}^T\boldsymbol{G}\boldsymbol{Z}, \;\; \boldsymbol{\Sigma}_{bb} = \boldsymbol{Z}^T\boldsymbol{Z},$$

since we have that $\boldsymbol{G} = \boldsymbol{G}^T$ and $\boldsymbol{G}^2 = \boldsymbol{I}$. Using the result from CCA [90] we have the the CCA is equal to the sum of the singular values of $\boldsymbol{\Sigma}_{bb}^{-1}\boldsymbol{\Sigma}_{ba}C_{aa}^{-1}\boldsymbol{\Sigma}_{ab}$ which corresponds to $\mathrm{Tr}((\boldsymbol{Z}^T\boldsymbol{Z})^{-1}\boldsymbol{Z}^T\boldsymbol{G}\boldsymbol{Z})$ since $\boldsymbol{\Sigma}_{ab} = \boldsymbol{\Sigma}_{ba}$ and $\boldsymbol{\Sigma}_{aa} = \boldsymbol{\Sigma}_{bb}$. First, notice that the CCA optimum solves the following eigenvalue problem

$$\boldsymbol{\Sigma}_{aa}^{-1}\boldsymbol{\Sigma}_{ab}\boldsymbol{\Sigma}_{bb}^{-1}\boldsymbol{\Sigma}_{ba}\boldsymbol{u}_{\mathrm{CCA}} = \lambda_{\mathrm{CCA}}\boldsymbol{u}_{\mathrm{CCA}},$$

but noticing that in our case $\boldsymbol{X}_{\mathrm{l}}$ and $\boldsymbol{X}_{\mathrm{r}}$ are permutations of each other, we directly obtain that

$$\boldsymbol{\Sigma}_{aa}^{-1}\boldsymbol{\Sigma}_{ab}\boldsymbol{\Sigma}_{bb}^{-1}\boldsymbol{\Sigma}_{ba}\boldsymbol{u}_{\mathrm{CCA}} = \lambda_{\mathrm{CCA}}\boldsymbol{u}_{\mathrm{CCA}} \iff \boldsymbol{\Sigma}_t^{-1}\boldsymbol{\Sigma}_b\boldsymbol{\Sigma}_t^{-1}\boldsymbol{\Sigma}_b\boldsymbol{u}_{\mathrm{CCA}} = \lambda_{\mathrm{CCA}}\boldsymbol{u}_{\mathrm{CCA}},$$

using the total, between and within-class covariance matrices. Now, starting from the LDA loss, we will see that we recover the CCA one thanks to the above facts

$$\boldsymbol{\Sigma}_b\boldsymbol{u}_{\mathrm{LDA}} = \lambda_{\mathrm{LDA}}\boldsymbol{\Sigma}_w\boldsymbol{u}_{\mathrm{LDA}} \qquad \text{(original LDA parameter)}$$

$$\iff \boldsymbol{\Sigma}_b\boldsymbol{u}_{\mathrm{LDA}} = \lambda_{\mathrm{LDA}}(\boldsymbol{\Sigma}_t - \boldsymbol{\Sigma}_b)\boldsymbol{u}_{\mathrm{LDA}}$$

$$\iff \boldsymbol{\Sigma}_b(1 + \lambda_{\mathrm{LDA}})\boldsymbol{u}_{\mathrm{LDA}} = \lambda_{\mathrm{LDA}}\boldsymbol{\Sigma}_t\boldsymbol{u}_{\mathrm{LDA}}$$

$$\iff \boldsymbol{\Sigma}_b\boldsymbol{u}_{\mathrm{LDA}} = \frac{\lambda_{\mathrm{LDA}}}{(1 + \lambda_{\mathrm{LDA}})}\boldsymbol{\Sigma}_t\boldsymbol{u}_{\mathrm{LDA}}$$

$$\iff \boldsymbol{\Sigma}_t^{-1}\boldsymbol{\Sigma}_b\boldsymbol{u}_{\mathrm{LDA}} = \frac{\lambda_{\mathrm{LDA}}}{(1 + \lambda_{\mathrm{LDA}})}\boldsymbol{u}_{\mathrm{LDA}}$$

$$\iff \boldsymbol{\Sigma}_t^{-1}\boldsymbol{\Sigma}_b\boldsymbol{\Sigma}_t^{-1}\boldsymbol{\Sigma}_b\boldsymbol{u}_{\mathrm{LDA}} = \left(\frac{\lambda_{\mathrm{LDA}}}{1 + \lambda_{\mathrm{LDA}}}\right)^2\boldsymbol{u}_{\mathrm{LDA}} \qquad \text{(original CCA problem)}$$

hence given the LDA or CCA solutions one can recover the others, the only difference lies in the singular values, which can also be found from each other. Also, recall that the eigenvalues are real positive as per lemma 1 ensuring the well posed division used above.

### F.16    Proof of VICReg Optimality (Theorem 9)

The proof first uses Theorem 1 that demonstrates how the optimal representation $\boldsymbol{Z}$ will have the left singular vectors of $\boldsymbol{G}$ up to rotations for the ones with multiplicity greater than one. Then, using that, we obtain from Theorem 9 that if those singular vectors are able to span the left singular vectors of $\boldsymbol{Y}$ then there exists a linear probe that will solve the task at hand, giving the desired result.