# OpenReview forum: "Contrastive and Non-Contrastive Self-Supervised Learning Recover Global and Local Spectral Embedding Methods"
_NeurIPS.cc/2022/Conference — NeurIPS 2022 Accept_

### Official Review · Reviewer_jF2D · 2022-07-06

**Rating:** 6
**Confidence:** 4
**Soundness:** 4 excellent
**Presentation:** 4 excellent
**Contribution:** 3 good

**Summary:**

This paper proposed a unfied framework for self-supervised learning with the help of spectral manifold learning .the authors show that VICReg, SimCLR, BarlowTwins are specical cases of their proposed framwork. They points out the prerequisite about the production of optimal self-supervised representations for downstream tasks.

**Questions:**

1. how the loss of DINO can be reformulated into the way of Eq. (5)?
1. can the sprectral manifold learning be applied on DINO so as to unify it into your proposed framework?
if the authors can address my concerns, I will raise my rate.

**Ethics Review Area:**

["I don’t know"]

**Strengths And Weaknesses:**

Strengths
1. this paper is well written and easy to follow
2. the idea of specral manifold learning for unifying SimCLR(NNCLR), BarlowTwins and VICReg seems to interesting for me
3. this paper is technically sound

Weakness
1. I think the authors overclaim their contributions of this paper. From my view, they have only consider VICReg, BarlowTwins and SimCLR in this paper. Although They have mentioned DINO in this paper, they do not reveal the connection between DINO and others.

---

> ### Author Response · Authors · 2022-08-02
> **Answer to Reviewer jF2D**
>
> We thank the reviewer for their insightful comments and suggestions. We have provided below our answers to each of the reviewer's comments. We will be happy to continue answering further question during the discussion period.
>
> **Reviewer**: *how the loss of DINO can be reformulated into the way of Eq. (5)?*
>
> **Answer**: We apologize for the confusion around DINO (and SimSIAM/BYOL). In the original manuscript we mentioned that DINO was equivalent to SimSIAM but where the embedding space is not continuous but quantized/discrete. Hence going from the cosine similarity to a cross-entropy based metric. However this connection was indeed quite loose. Instead, we have now simply removed any mention of DINO and BYOL, and instead explicitly cited an existing work that relates SimSIAM/BYOL to BarlowTwins (in Fig. 1). We thus leave the reader to consult that reference for further details on this connection to avoid any confusion in our submission.
>
> **Reviewer**: *can the sprectral manifold learning be applied on DINO so as to unify it into your proposed framework?*
>
> **Answer**: Yes, the study proposed by [4] (that we have now emphasized in Fig. 1) does just that connecting methods like SimSIAM and BYOL to different variants of Deep CCA which is itself equivalent to BarlowTwins. We have correctly added this connection in the paper thanks to your suggestion. In addition of this change we have recently found yet another SSL method (W-MSE from [5]) which corresponds to the particular “constrained VICReg” that we studied theoretically in our submission. Hence with the addition of those variants we believe that our study now covers most of the existing SSL variants. We hope that those additional results answer the reviewer’s concerns and we are happy to keep iterating on those changes during the discussion process.
>
>
> [4] Lee, Lei, Saunshi, Zhuo, 2021. Predicting What You Already Know Helps: Provable Self-Supervised Learning.
>
> [5] Ermolov, Aleksandr, et al. "Whitening for self-supervised representation learning." International Conference on Machine Learning. PMLR, 2021.

---

### Official Review · Reviewer_ZYB8 · 2022-07-09

**Rating:** 6
**Confidence:** 3
**Soundness:** 3 good
**Presentation:** 3 good
**Contribution:** 2 fair

**Summary:**

This paper explores the links between self-supervised losses, metric learning and spectral embedding methods. It specifically investigates which representations are learnt when the embedding is chosen to be linear, and carefully investigates how the choice of loss affects their rank.

**Questions:**

**Clarification**
1. Figure 3 is intruiging. Indeed, VIC is composed of three terms: the "variance" and "covariance" whiten the representation, while the "invariance" term incorporates "downstream" task information (e.g. an image and its rotation should have the same label). However, Figure 3 shows that the downstream task can be perfectly when the "invariance" term barely intervenes (gamma = 0.001, just enough to make Z full rank). In other words, this suggests that a (nonlinear) PCA or ICA would be enough to solve a downstream task, which is counter-intuitive to SSL which posits that injecting downstream task information via pairwise relations (matrix G) is essential. Could the authors comment on that?
2. The spectral analysis VICReg and Barlow Twins are presented as separate contributions, but isn't Barlow Twins subsumed by VICReg (setting the 'variance' term to zero)?
3. The authors reference "BarlowTwins, DINO, BYOL, and SimSIAM" but DINO and BYOL do not seem to appear subsequently in the analysis: could the authors clarify?
4. It is not very clear to me how equation (4) is built and subsequently used.
5. "The authors write "VICReg" recovers [...] Linear Discriminant Analysis (LDA)": can the authors specify is that is when the whitening is enforced (Theorem 2) or relaxed with alpha and beta (equation 1)?
6. The sentence "This already brings a contrast from non-contrastive methods showing that contrastive methods learn to produce signals such that the graph estimate is close to the known graph" seems like it is over-reaching. I am not sure there is a clear definition of "contrastive" and "non-contrastive" methods in the first place: it could be possible after all to reformulate VICReg as a contrastive loss with implicit negative samples. Also, SimCLR learns a graph estimate (G_hat in equation 15) while VICReg does not (equation 1), but that is a choice in the loss. Perhaps it is clearer to compare VICReg with SimCLR directly, without extrapolating to "non-contrastive" vs. "contrastive"?
7. [1] reformulates PCA and Kernel PCA using a Laplacian given in Section 3.3. "Principal Component Analysis", matrix Lij. This corresponds to solving the "variance" and "covariance" terms of VICReg in this submission, as they are enforced in Theorem 2. Could the authors comment on that? Does that connect with the authors' equation 9 which combines the Laplacians of the PCA terms ("variance" and "covariance") with the Laplacian of the downstream task ("invariance" to G)?




 **Minor Typos or Formulations**
- "We can directly remind the VICReg" -> recall
- "variosu" -> various
- "there exists many local minimum" -> there exist many local minima
- "to solve the task at hand perfect" -> perfectly
- "Prior to moving" -> Before moving
- "unify SSL methods under the realm" -> under the helm
- "similar pairwise distance (usually l2) than" -> similar [...] to
- "we ought ot" -> to
- "poitn" -> point


**Related Work**

I believe there is very relevant work that could be referenced by the authors.
- [1] unifies many embedding methods (e.g. PCA, spectral methods, MDS, UMAP, Isomap) under the helm of pairwise metric learning
- [2] already draws a link between SiamSiam and nonlinear CCA
- [3] connects spectral methods (over a Kernel, e.g. Covariance or Graph Laplacian) with SSL objectives, and possibly provides theoretical justification for tricks of the trade (e.g. 'stop-gradient' for so-called 'Non-Contrastive' SSL to avoid collapse).


Abbreviation: SSL = Self-Supervised Learning.

[1] Agrawal, Ali, Boyd, 2021. Minimum Distortion Embedding.

[2] Lee, Lei, Saunshi, Zhuo, 2021. Predicting What You Already Know Helps: Provable Self-Supervised Learning.

[3] Pfau, Petersen, Agarwal, Barrett, Stachenfeld, 2019. Spectral Inference Networks: Unifying Deep and Spectral Learning.


**Ethics Review Area:**

["I don’t know"]

**Limitations:**

I have no potential negative societal impact to raise.

**Strengths And Weaknesses:**

**Strengths**

Though many connections between metric learning, SSL and spectral embedding methods have been previously explored [1, 2, 3], this paper is a welcome addition. Explicitly writing out the optimal representations in the linear regime is useful. Explicitly showing how the choice of SSL loss impacts the rank of the learnt representation (in comparison of the rank of the downstream information encoded in G) is  a welcome contribution to recent investigations on the "dimensional collapse" of SSL representations.


**Weaknesses: better referencing known terminology and problems**

Given the authors are connecting SSL to known Spectral Embedding methods, it would add significant clarity to draw on known terminology as well. For example, VICReg is composed of three terms that are eponymously described as 'variance-invariance-covariance' with a rather new vocabulary of 'dimensional collapse' and 'semantic' similarity. On a simpler level, the VICReg loss is simply Robust Nonlinear PCA, where Terms 1 and 2 'whitens the signal' and Term 3 encourages robustness to perturbations (e.g. jitter or rotation) encoded in G. Specifically,
- Term 1 removes the scaling ambiguity of PCA (by setting the variances to 1)
- Term 2 encourages orthogonal representations
- Term 3 encourages robustness to user-specified transforms (e.g. rotation)

Relating this to known terminology (PCA) is useful because it *explains* the observations made by the authors, when they say
- "the optimum is not unique" or "there exist many local minima": this is due to well-known indeterminacies for PCA (e.g. permutation, offset) and is not a new problem
- Fig 2, gamma = 0 vs. gamma = 0.1 is essentially PCA vs. Robust PCA. Expectedly, PCA has known indeterminacies (Panel 3) and Robust PCA blurs the landscape (Panel 3) and smoothens the latent space (Panel 2) as it desensitizes the loss to local perturbations.

It seems like these observations are framed as a new problem of VICReg when they in fact inherit from PCA.


Abbreviation: SSL = Self-Supervised Learning.

[1] Agrawal, Ali, Boyd, 2021. Minimum Distortion Embedding.

[2] Lee, Lei, Saunshi, Zhuo, 2021. Predicting What You Already Know Helps: Provable Self-Supervised Learning.

[3] Pfau, Petersen, Agarwal, Barrett, Stachenfeld, 2019. Spectral Inference Networks: Unifying Deep and Spectral Learning.

---

> ### Author Response · Authors · 2022-08-02
> **Answer to Reviewer ZYB8 (part 1 of 2)**
>
> We thank the reviewer for their insightful comments and suggestions. We have provided below our answers to each of the reviewer's comments. We will be happy to continue answering further question during the discussion period.
>
> **Reviewer**: *Figure 3 is intruiging. Indeed, VIC is composed of three terms: the "variance" and "covariance" whiten the representation, while the "invariance" term incorporates "downstream" task information (e.g. an image and its rotation should have the same label). However, Figure 3 shows that the downstream task can be perfectly when the "invariance" term barely intervenes (gamma = 0.001, just enough to make Z full rank). In other words, this suggests that a (nonlinear) PCA or ICA would be enough to solve a downstream task, which is counter-intuitive to SSL which posits that injecting downstream task information via pairwise relations (matrix G) is essential. Could the authors comment on that?*
>
> **Answer**: This is a very interesting observation that we have also made ourselves in light of this figure. This is a potential reason why VICReg might provide slightly better downstream performance as opposed to e.g. SimCLR: by controlling the losses independently, one can reach a full-rank representation that is only slightly tilted towards the invariants prescribed by the employed data-augmentation and G used to generate the positive views. Hence, in addition to being optimal to the task at hand (prescribed by the augmentation) it is also the most informative for other downstream tasks, as opposed to collapsed representations. We should highlight though that this impressive gap occurs at a small *but nonzero* gamma. And in fact, at gamma=0 (see left-most part of the plot), the performance is near random, suggesting that a pure nonlinear PCA/ICA would most likely fail too, as already witnessed many years ago when nonlinear PCA was employed to learn representations of high-dimensional data. We believe that this finding sheds new light for future works e.g. to automatically select an optimal gamma parameter that we hope to study further in subsequent works.
>
> **Reviewer**: *The spectral analysis VICReg and Barlow Twins are presented as separate contributions, but isn't Barlow Twins subsumed by VICReg (setting the 'variance' term to zero)?*
>
> **Answer**: This is a good point raised by the reviewer which is entirely correct. We proposed a separate section/contribution for both since as far as we know, previous empirical/theoretical studies of SSL consider those methods as two separate ones. Additionally, although in their current form (with two views), an informed reader might easily go from VICReg to BarlowTwins and back, this link isn’t so clear for multi-view SSL (with V>2 views). In fact, VICReg easily deals with this setting by keeping the same variance+covariance terms but performing the invariance over all positive pairs or between their mean and each view (both being equivalent mathematically). However such an extension for BarlowTwins (CCA) seems less “trivial” e.g. see the many multi-view CCA variants that have emerged [1,2,3]. For that reason, we opted to present VICReg and BarlowTwins as two separate contributions and demonstrated in a few particular settings how both methods fall back to performing the same transformations (e.g. Sec. 5.2), hence hinting at the reviewer’s observation.
>
> [1] Rupnik, Jan, and John Shawe-Taylor. "Multi-view canonical correlation analysis." Conference on data mining and data warehouses (SiKDD 2010). 2010.
>
> [2] Guo, Chenfeng, and Dongrui Wu. "Canonical correlation analysis (CCA) based multi-view learning: An overview." arXiv preprint arXiv:1907.01693 (2019).
>
> [3] Chen, Jia, Gang Wang, and Georgios B. Giannakis. "Multiview canonical correlation analysis over graphs." ICASSP 2019-2019 IEEE International Conference on Acoustics, Speech and Signal Processing (ICASSP). IEEE, 2019.

---

> > ### Comment · Reviewer_ZYB8 · 2022-08-08
> > **Answer to the authors**
> >
> > I appreciate the authors' answers which help reframe these questions in context. Apologies for the delay.
> >
> > Overall, I believe that connecting Self-Supervised methods to a well-established literature (Spectral Representation Learning) should help us understand them better. And while it is the case mathematically, I think it does not easily transpire in the text. I appreciate that unifying methods with common notations leads to exhaustive formulae and terminology, and part of that is unavoidable. But more clarity could come from interpreting the graphs (the comments are quite descriptive) and saying in text what the different hyperparameter configurations mean. For example,
> >
> > - Recapping more clearly that the VICReg loss structures the representation in two ways: it "whitens" it (prior structure) while incorporating symmetries we anticipate (downstream task structure). The prior structure is controlled by alpha, beta; the downstream structure is controlled by gamma.
> >
> > Having said that, the big question in SSL is: how does incorporating downstream task information (via gamma) change the representation?
> >
> > - Figure 2: for both gamma = 0 vs. gamma = 0.1, the representation should be "whitened". So we're actually isolating the marginal effect of incorporating downstream structure via gamma. This is done via the invariance (e.g. to rotations) term. We expect that this creates blur as more data points (e.g. an image and its rotations) are mapped to a same representation. That is what we see, along with global optima which come from the indeterminacies you described [Section 3.1., Optimal Representation paragraph].
> >
> > - Fig 3: again, the difference between gamma = 0 vs. gamma > 0 is the marginal effect of incorporating downstream structure. Small gamma (1e-3) suggests the "whitened" representation requires "just enough" downstream task to be non-degenerate (full rank). This tells us that **injecting downstream information is a threshold effect** (gamma = 0 vs. != 0), rather than a continuous effect: that is perhaps unexpected. This also tells us that a "whitened representation" is **almost good enough**, which is interesting as it does not require information from the downstream task to (pre)train.
> >
> > I think these interpretations to your graphs are essential, especially if the end goal of formalizing these connections is to answer questions about Self-Supervised Representation Learning. I think it helps the reader understand these conclusions and lessens the cognitive load of doing the back-and-forth between the figure and the main text to remember (i) what each hyperparameter (alpha-beta-gamma) meant (ii) how each term (variance-covariance + invariance) is interpreted (prior structure + downstream structure).
> >
> > I do not know if this is just my personal take on this but I hope this can be constructive.
> >
> > After reading the changes to the manuscript, as well as the answers and comments listed here, I too will opt to maintain my score.

---

> > > ### Author Response · Authors · 2022-08-08
> > > **Answer to Reviewer ZYB8**
> > >
> > > We thank the reviewer for reviewing our answer and for going through the revised version of our submission.
> > >
> > > First, we would like to mention to the reviewer that we might have missed the original reviewer's point asking for a more explicit interpretation of our results. **We thus have submitted a new revision of the supplementary materials where in the first page, we summarize precisely the found insights obtained from studying SSL methods from a spectral embedding viewpoint**. In the event of acceptance, this summary will me move to the conclusion section to ensure that our results are stated as clearly as possible. We thank the reviewer for reinforcing their point in this latest comment and hope that this last revision satisfies the reviewer.
> > >
> > > Regarding your other point:
> > >
> > > - in Figures 2 and 3, we are employing VICReg (and not the constrained VICReg i.e. W-MSE). Hence, depending on the dimension of the representation and rank of G, having a whitened representation is not always guaranteed for nonzero gamma, however, this was true in our settings (high classification performance was obtained by tilting the whitened representation to the invariant space prescribed by G). We will be sure to state this in the caption of those figures. That being clarified, we entirely agree with the reviewer's point: **a whitened representation is almost good enough**. In fact, we would go one step-further (also based on the W-MSE/constrained VICReg paper's experiments) and claim that **a whitened representation is good enough**, and that the challenge is to find **which whitened representation**. For example on Imagenet, the input space representation is 3x256x256=196608 (common resized shape). But the representation is at most 2048 (this is for a resnet50, other models tend to have slightly smaller dimensions). Hence what we hope to have conveyed in our study is that (in general) *SSL aims at finding the correct low-dimensional whitened representation from very-high dimensional data and partial observation of that representation's kernel space (from the positive samples)*.  From that statement, the bridge to spectral embedding naturally emerges. If the reviewer finds the above insightful and aligned with their comment we would be happy to further add such a note in the conclusion of the final version of our submission.

---

> ### Author Response · Authors · 2022-08-02
> **Answer to Reviewer ZYB8 (part 2 of 2)**
>
> **Reviewer**: *The authors reference "BarlowTwins, DINO, BYOL, and SimSIAM" but DINO and BYOL do not seem to appear subsequently in the analysis: could the authors clarify?*
>
> **Answer**: We apologize for the confusion. As already known by the reviewer (see e.g. [4] from the suggested references) BarlowTwins and SimSIAM/BYOL are already closely related. From that, DINO is simply a different metric space than SimSIAM (quantization step and cross-entropy metric as opposed to continuous output and euclidean metrics as was briefly mentioned at the beginning of Sec. 5.1). However, we do agree that this connection is light, and we thus removed any mention of DINO and BYOL in the manuscript. We instead emphasized in Fig. 1 the link between BarlowTwins and SimSIAM and referred the reader to the corresponding reference for further details.
>
> [4] Lee, Lei, Saunshi, Zhuo, 2021. Predicting What You Already Know Helps: Provable Self-Supervised Learning.
>
> **Reviewer**: *It is not very clear to me how equation (4) is built and subsequently used.*
>
> **Answer**: Those equations were provided for completeness to follow our notations and to describe precisely how one builds the left/right views only from the data matrix and possibly G. However, as correctly pointed out by the reviewer, the formulation was overly verbose and has thus been updated to be much more concise and to the point. We hope that this clarifies this part.
>
> **Reviewer**: *The authors write "VICReg" recovers [...] Linear Discriminant Analysis (LDA)": can the authors specify is that is when the whitening is enforced (Theorem 2) or relaxed with alpha and beta (equation 1)?*
>
> **Answer**: This is a miss from our side. We have now precisely stated that this occurs for the constrained VICReg i.e. when whitening is enforced.
>
> **Reviewer**: *The sentence "This already brings a contrast from non-contrastive methods showing that contrastive methods learn to produce signals such that the graph estimate is close to the known graph" seems like it is overreaching. I am not sure there is a clear definition of "contrastive" and "non-contrastive" methods in the first place: it could be possible after all to reformulate VICReg as a contrastive loss with implicit negative samples. Also, SimCLR learns a graph estimate (G_hat in equation 15) while VICReg does not (equation 1), but that is a choice in the loss. Perhaps it is clearer to compare VICReg with SimCLR directly, without extrapolating to "non-contrastive" vs. "contrastive"?*
>
> **Answer**: This is a great point raised by the reviewer. It is indeed unclear what is the precise definition of a contrastive and a non-contrastive method. As pointed out, one could consider VICReg to be contrastive since as we pointed out in Eq. 7, the variance+covariance term implicitly makes all the samples in the dataset to be pushing each other away akin to negative pairs. We thus corrected this and simply stated in our manuscript that “this is a distinction between SimCLR and VICReg”, as well suggested. We also reduced the strength of our wording in the conclusion of the revised version to avoid directly comparing contrastive and non-contrastive methods explicitly
>
> **Reviewer**: *[1] reformulates PCA and Kernel PCA using a Laplacian given in Section 3.3. "Principal Component Analysis", matrix Lij. This corresponds to solving the "variance" and "covariance" terms of VICReg in this submission, as they are enforced in Theorem 2. Could the authors comment on that? Does that connect with the authors' equation 9 which combines the Laplacians of the PCA terms ("variance" and "covariance") with the Laplacian of the downstream task ("invariance" to G)?*
>
> **Answer**: This is an interesting observation. It is indeed true that various spectral methods can fall back to generalizations of kernel PCA. In fact, with the proper choice of kernel, the eigendecomposition of the latter can be made to recover Laplacian Eigenmaps, Isomaps and the likes. We have made sure to include this reference and the others suggested by the reviewers in Sec. 3.2. We also added yet another reference [5]  that we originally missed and that we recently came across, that already employed the constrained VICReg loss i.e. they impose an exact embedding whitening and then minimize the invariance term. This allows us to extend our analysis to yet another SSL loss and to include the corresponding reference which is crucial as [5] did not provide any analysis of the proposed “constrained VICReg loss”.
>
> [5] Ermolov, Aleksandr, et al. "Whitening for self-supervised representation learning." International Conference on Machine Learning. PMLR, 2021.
>
>
>
>
> We thank the reviewer for pointing out typos in the manuscripts, we have taken care of them

---

### Official Review · Reviewer_VtRd · 2022-07-11

**Rating:** 6
**Confidence:** 3
**Soundness:** 3 good
**Presentation:** 2 fair
**Contribution:** 3 good

**Summary:**

This paper introduces a general framework that leverages spectral methods to unify the representative self-supervised learning (SSL) methods such as VICReg, SimCLR, BarlowTwins. The authors provide a detailed theoretical analysis of different SSL frameworks and demonstrate the properties of their learned representations.

**Questions:**

I am generally satisfied with the paper, but have an open question based on the analysis:

As these methods have different properties, while some of them are complementary, e.g., VICReg prevents dimensional collapse, while SimCLR could have a dimensional collapse, is there any way to propose a solution that complements each other's advantages?

**Limitations:**

The authors have discussed the limitation of the paper.

**Strengths And Weaknesses:**

### Strength:

Unifying existing self-supervised learning methods with spectral methods looks interesting. It also provides new insights into the community to help understand how SSL works. Both the motivation and the technical details in the experiment designs seem sound. The authors have cited most of the relevant papers and well summarized previous works.

### Weakness:

Although the paper provides an insightful view of understanding the mechanism of SSL, most of the analysis is only valid in the pre-assumptions, some of which might have a gap in practical usage. For example, the properties of VICReg are based on the linear network assumption, while in practice, the encoder backbones are often non-linear. As the authors claim this paper aims to provide some guidelines to practitioners, these gaps could limit their insights to them.

The example in Eq (4) and (5) is a little bit hard to follow. It is not obvious see the logical relation between these two formulations.

The presentation quality is fairly good but needs further polishing. It feels like the paper is finished in a rush, and some paragraphs are not clear to the readers. Some abbreviations are not explained with the full name or corresponding citation when first appearing in the text, e.g., DA, DN, which might confuse the readers.

The experiment/simulation settings are not specified in the paper, although most validations are conducted on toy examples. For example, what datasets are used for these proof of concepts?

Most of the discussions are based on the assumption that the downstream task is classification, while in practice, SSL methods are widely used in various settings, including detection, segmentation, generative modeling, etc. The insight to practitioners might be limited in the scope of the classification.

Minors:

The notation $h$ in Eq (3) is not used; simCLR -> SimCLR

Line 91: loose -> lose;

Line 98: variosu -> various

---

> ### Author Response · Authors · 2022-08-02
> **Answer to Reviewer VtRd**
>
> We thank the reviewer for their insightful comments and suggestions:
>
>
> **Reviewer**: *Although the paper provides an insightful view of understanding the mechanism of SSL, most of the analysis is only valid in the pre-assumptions, some of which might have a gap in practical usage.*
>
> **Answer**: We thank the reviewer for pointing out this point which was due to an unclear introduction from our part. In short, all our results hold in two settings: the linear setting and the non-parametric/infinite capacity regime. Those two special cases cover the two ends of the spectrum of models one could employ in practice. A Deep Network would be in-between with high capacity but also with an implicit bias from the choice of architecture. We have made sure to point this out precisely in the intro and as one of the limitation of the current study in the conclusion.
>
> **Reviewer**: *The example in Eq (4) and (5) is a little bit hard to follow. It is not obvious see the logical relation between these two formulations.*
>
> **Answer**: Those equations were provided to describe precisely how one builds the left/right views only from the data matrix and possibly G. However, as correctly pointed out by the reviewer, the formulation was overly verbose and has thus been updated to be much more concise and to the point. We hope that this clarifies this part.
>
> **Reviewer**: *The presentation quality is fairly good but needs further polishing. It feels like the paper is finished in a rush, and some paragraphs are not clear to the readers. Some abbreviations are not explained with the full name or corresponding citation when first appearing in the text, e.g., DA, DN, which might confuse the readers.*
>
> We apologize for missing out those notations and for the writing feeling rushed. We hope that the numerous results and connections we obtained convinced the reviewer that a lot of work has been put into this submission. We will be sure to carefully clarify and improve the English quality of the paper for the final version, and we have added in the revised manuscript a better formalism around each abbreviation and have made sure that they are well introduced. We have also slightly reworked a few parts of the paper that were found to be problematic. We recall that all the edits are highlighted in blue.
>
> **Reviewer**: *The experiment/simulation settings are not specified in the paper, although most validations are conducted on toy examples. For example, what datasets are used for these proof of concepts?*
>
> **Answer**: We have added the experiment details in a new section (Appendix A) and we will be sure that all the figures are easily reproducible from the codebase that will be released upon final decision
>
> **Reviewer**: *Most of the discussions are based on the assumption that the downstream task is classification, while in practice, SSL methods are widely used in various settings, including detection, segmentation, generative modeling, etc. The insight to practitioners might be limited in the scope of the classification.*
>
> **Answer**: This is a very good point that we forgot to mention in our submission. Although it is common in many cases to turn various downstream tasks into classification ones e.g. by quantizing the target outputs, we have made sure to be precise in the abstract/introduction stating explicitly that all our insights on downstream task performance results are specifically for classification tasks. We hope that this clarification will allow the reader to exactly pinpoint the results that we are providing.
>
> **Reviewer**: *As these methods have different properties, while some of them are complementary, e.g., VICReg prevents dimensional collapse, while SimCLR could have a dimensional collapse, is there any way to propose a solution that complements each other's advantages?*
>
> **Answer**: This is a very interesting direction that we hope to explore as part of future work. In light of our finding, we can already provide a few simple routes that would allow us to combat SimCLR’s dimensional collapse. A first one would be to exploit the properties of G by carefully setting the batch size (B), number of views (V), and representation dimension (K) so that the optimal representation is already full-rank. As per our results, one simple strategy would be to set B=V*K. Another possibly more informed solution would be to combine both losses and in particular the SimCLR InfoNCE term and the variance+covariance terms of VICReg. This would allow for a more fine-grained control of “how much collapse does one want” and could provide very similar behavior as seen for VICReg. This however would come at the cost of introducing yet another hyper-parameter to tune. If the reviewer feels that our submission would benefit from adding this comment, we will be happy to add it as part of the conclusion or appendix in the final version of the submission.
>
>
> We thank the reviewer for pointing out typos in the manuscripts, we have taken care of them.

---

> > ### Comment · Reviewer_VtRd · 2022-08-08
> > **Response to authors**
> >
> > I appreciate the authors' efforts in carefully addressing my questions. After reading through all reviews and the responses, I prefer to keep my score and be inclined to vote an acceptance unless severe concerns were raised.

---

> > > ### Author Response · Authors · 2022-08-08
> > > **Answer to Reviewer VtRd**
> > >
> > > We thank Reviewer VtRd for going through our revision and comments. We are delighted to read that we addressed the reviewer's questions.
> > >
> > > We remain happy to answer any further questions that might occur to the reviewer before the end of the discussion period.

---

### Official Review · Reviewer_W5rv · 2022-07-16

**Rating:** 5
**Confidence:** 3
**Soundness:** 2 fair
**Presentation:** 2 fair
**Contribution:** 3 good

**Summary:**

This paper provides a theoretical study of the popular self-supervised learning (SSL) techniques VICReg, SimCLR and BarlowTwins. The main results are closed-form optimal representations for each method as a function of the training data and the sample-relation matrix (which indicates which samples are from the same class, e.g. this could be constructed as the matrix that indicates which samples are augmentations of the same image). The paper further provides simplified versions of these expressions in linear settings and uses them to show equivalence between the SSL methods and various spectral methods, and provides a study on downstream task performances.

**Questions:**

If the authors can provide additional explanation of Figures 2, 3, and 4 that would be helpful.

**Limitations:**

The limitations of the theory are not discussed, and an assumption is missing (see above). There are no potential negative societal impacts.

**Strengths And Weaknesses:**

Strengths

1. The paper studies the important problem of theoretically understanding increasingly popular SSL methods.

2. To my knowledge, no other work has derived such expressions concerning SSL optimal solutions, or made the connections between SSL and spectral embedding methods.

3. The scope of the paper is comprehensive: all of the most popular SSL techniques are analyzed. Clearly a lot of work went into writing this paper.

Weaknesses

1. It is difficult for me to understand the insights of this work because the writing is so dense. Often expressions are introduced without sufficient background and conclusions drawn without sufficient explanation and/or motivation. E.g. $\mathcal{L}\_{var}$ and $\mathcal{L}\_{cov}$ in Theorem 2 and equations (13) and (14) - intuitively how do these equations arise and why are they useful? The figures are especially information-overloaded and difficult to parse. The claims relating to downstream task performance are not clear. Overall, it should be abundantly clear what the key messages are from this paper, but they are lost in the density of the work.

2. There is insufficient discussion of other theoretical studies of SSL (only a brief mention in the introduction).

3. Some of the statements and proofs are not rigorous, as it is not formally defined what it means for one method to recover another. The proof of Theorem 3 requires an assumption that is not made in the theorem statement. Other proofs can use more formality (only English text is given and the reader is left to fill in the details) and the proof of Lemma 1 is missing.

---

> ### Author Response · Authors · 2022-08-02
> **Answer to Reviewer W5rv**
>
> We thank the reviewer for their insightful comments and suggestions. We have provided below our answers to each of the reviewer's comments. We will be happy to continue answering further question during the discussion period.
>
> **Reviewer**: *It is difficult for me to understand the insights of this work because the writing is so dense. Often expressions are introduced without sufficient background and conclusions drawn without sufficient explanation and/or motivation. E.g. Lvar and Lcov in Theorem 2 and equations (13) and (14) - intuitively how do these equations arise and why are they useful? The figures are especially information-overloaded and difficult to parse.*
>
> **Answer**: We apologize for missing the definitions of the Lvar, Lcov, Linv terms. We have now introduced them right after introducing the VICReg loss in the background section. We have also moved Eq (13) and (14) to the appendix E to allow for a discussion around them better highlighting the benefit of those derivations. In short, our main message was that “because we now have the general optimization form that recovers SimCLR as a special case when employing a specific regularizer, it is now possible to derive novel and alternative SSL methods by changing that regularizer e.g. as one would design to better suit an application at hand”. We have made that clear and pointed to other possible benefits in that new appendix E with reference in the main text (where (13) and (14) used to be).
>
> **Reviewer**: *The claims relating to downstream task performance are not clear. Overall, it should be abundantly clear what the key messages are from this paper, but they are lost in the density of the work.*
>
> **Answer**:The claim relating to downstream performances is in two steps. Our first (i) result ties many SSL methods to spectral embedding methods from which it is possible to characterize the spectral properties of the optimal representations that a model would learn. Given those properties, our second (ii) result is that it is then possible to characterize downstream task performance, especially with a linear prob and in a supervised/regression setting. While (i) was provided in the main text, (ii) was provided in the appendix as those are not our main contributions and mostly follow standard results e.g. from linear regression (but (i) is non-trivial and novel for the SSL community and the spectral embedding community). We have slightly updated our introduction to make the above clear. From (i) and (ii) though, we obtain one of our core observation: “if you have a good DA, SimCLR/BT/VICReg will all produce optimal representations for the downstream task” or conversely “regardless of the employed method (SimCLR/BT/VICReg) you can find a DA that will make your model learn an optimal representation for your downstream task”. In short, none of those methods is suboptimal in that regard. We hope that this message is clear from our introduction and are happy to further emphasize it in the final version of the paper.
>
> **Reviewer**: *There is insufficient discussion of other theoretical studies of SSL (only a brief mention in the introduction).*
>
> **Answer**: We have been sure to add additional references and further discussions, especially in Fig. 1 and after Thm 2. We will be happy to add further references if the reviewer feels that we are still missing preliminary works connection SSL and spectral embedding methods.
>
> **Reviewer**: *Some of the statements and proofs are not rigorous, as it is not formally defined what it means for one method to recover another. The proof of Theorem 3 requires an assumption that is not made in the theorem statement. Other proofs can use more formality (only English text is given and the reader is left to fill in the details) and the proof of Lemma 1 is missing.*
>
> **Answer**: We have added a formal proof of Lemma 1 below it, we apologize for forgetting to add it in the original manuscript, and we have also updated some of the original proofs to be more thorough and systematic, avoiding hand-wavy arguments (all changes are in blue). We also added the assumption needed in the Proof of Thm. 3 before Thm. 3’s statement in the main text to be sure that no assumption is missing from the theorems’ statements and from the main text.

---

> > ### Comment · Reviewer_W5rv · 2022-08-08
> > **Response**
> >
> > I thank the reviewers for making appropriate revisions and their helpful response in clarifying the contributions of the paper. I have raised my score accordingly.

---

> > > ### Author Response · Authors · 2022-08-08
> > > **Answer to Reviewer W5rv**
> > >
> > > We are grateful to Reviewer W5rv for not only acknowledging our revision and answer, but also for finding our updates satisfactory. We strongly believe that those suggested changes have made our submission stronger.
> > >
> > > We remain available to answer any additional comment/question that the reviewer might have before the end of the discussion period.

---

### Author Response · Authors · 2022-08-02
**General answer to all reviewers**

We thank all the reviewers for their insightful reviews that allowed us to greatly improve our submission. By combining the comments and suggestions from the reviewers we have been able to connect additional SSL methods to spectral embedding methods, to clarify our claims, proofs and limitations, and to tilt the paper to be as insightful as possible for the community.

We provide reviewer specific answers and simply remind the reviewers here that all the codebase to reproduce each of the manuscript figure will be released publicly, and that in the event of acceptance, the extra page allowed for the camera ready version will allow us to strengthen the descriptions of Fig 2, 3, 4 and to further strengthen the insights obtained from our analysis. We already performed edits to that end in the revised manuscript which are highlighted in blue.

We hope that those edits and our answers cover all the reviewers’ points, if not we will be happy to keep working on them during the discussion period.

---

### Meta-Review · Area_Chair_zD6t · 2022-08-26

**Recommendation:** Accept
**Confidence:** Certain

**Metareview:**

This paper focuses on providing some theoretical intuition/understandings of popular self-supervised learning (SSL) methods. The authors develop closed-form optimal representations for various method as a function of the training data and the sample-relation matrix. The authors also provide further intuition by developing simplified versions of these expressions in linear settings which they use to show an equivalence of sorts between SSL and various spectral methods and how it affects downstream tasks. Overall the reviewers were positive and thought the paper had nice insights. They did raise some concerns about the quality of exposition and various detailed technical issues. Most of the technical issues seems to have been addressed by the authors in their response. I concur with the reviewers. The paper has nice insights and therefore I recommend acceptance. I do however recommend that the authors further polish the paper for the camera ready version by addressing the issues raised by the reviewers especially about the exposition.

**Award:**

No

---

### Decision · Program_Chairs · 2022-09-14

Accept